# Ancient recycled lower crust in the mantle source of recent Italian magmatism

Janne M. Koornneef [1], Igor Nikogosian [1,2], Manfred J. van Bergen[2], Pieter Z. Vroon[1] & Gareth R. Davies [1]

Recycling of Earth's crust through subduction and delamination contributes to mantle heterogeneity. Melt inclusions in early crystallised magmatic minerals record greater geochemical variability than host lavas and more fully reflect the heterogeneity of magma sources. To date, use of multiple isotope systems on small (< 300 μm) melt inclusions was hampered by analytical limitations. Here we report the first coupled Sr-Nd-Pb isotope data on individual melt inclusions from potassium-rich lavas from neighbouring Quaternary volcanoes in central Italy and infer the presence of a previously unidentified ancient lower crustal component in the mantle. We suggest derivation from Variscan or older basement included in the upper mantle by either delamination, sediment recycling, subduction erosion and/or slab detachment processes during Cenozoic subduction and collision of the western Mediterranean. The capability to determine isotope ratios in individual melt inclusions permits the detection of distinctive mantle contaminants and can provide insights into how geodynamic processes affect subduction recycling.

[1] Vrije Universiteit Amsterdam, De Boelelaan 1085, 1081 HV Amsterdam, The Netherlands. [2] Utrecht University, Budapestlaan 4, 3584 CD Utrecht, The Netherlands. Correspondence and requests for materials should be addressed to J.M.K. (email: j.m.koornneef@vu.nl)

Knowledge of the magnitude of mantle heterogeneity is crucial to understand the tectono-magmatic processes that govern fluxes in and out of the mantle and how these have changed the composition of this prominent Earth reservoir with time[1]. Growing evidence suggests that isotopic compositions of bulk lavas record signatures from a heterogeneous mantle resulting from the introduction of crustal materials by subduction and/or delamination throughout Earth's existence[2]. Bulk magmas are, after extraction from the mantle, however, subject to assimilation and homogenisation within the crust, concealing the true extent of mantle heterogeneity. In contrast, melt inclusions (MIs), pockets of melts trapped within crystals growing at depth, are shielded from subsequent mixing and contamination of the magmas on their way to the surface, and may thus provide a more direct picture of the composition of the mantle below the crust[3–6]. Primitive olivine-hosted melt inclusions trapped at the earliest stages of magma evolution can be distinguished from those that represent more evolved melt and were affected by assimilation in the crust, based on the major element composition of the host olivine (e.g., high forterite Fo mol% = Mg/(Mg + Fe) × 100) and the major and trace-element compositions of the MIs. Isotopic analyses of primitive melt inclusions, used to evaluate mantle heterogeneity, are, however, challenging as they are usually <300 μm in size providing limited amounts of the elements of interest (Sr, Nd, Pb)[7].

Large diversity in mantle compositions is to be expected at converging plate boundaries involving subduction and collision of old passive continental margins, due to the gradual changes in subducted materials (e.g., from oceanic to continent-derived) and possible thermo-mechanical erosion effects induced by slab tear and break-off. Geochemical and isotopic diversity will be especially marked when converging plates include old continental crust as subduction-related processes such as sediment and fluid input, subduction erosion, and lithosphere delamination will include material of markedly different age and isotopic compositions. Peninsular Italy, located within the Central-Western Mediterranean region, is a prime example of such a dynamically evolving collision setting (Fig. 1). Nearby crustal basement segments record ages up to Precambrian (e.g., Massif Central, Sardinia, Corsica, Calabria, Adria) reflecting formation on the Northern margin of Gondwana in the Early Palaeozoic, drift and amalgamation of micro-continental slivers onto Laurussia and subsequent reworking during the Variscan and Alpine orogenies[8–12]. The current geodynamic setting of Italy is the result of active westward subduction of the Adriatic plate that started in the late Eocene and was accompanied by eastward roll-back until the plate boundary reached the Dinaride continental lithosphere[13,14]. Subduction has now ceased, with the exception of the Calabrian Arc sector. Abundant Quaternary volcanism on mainland Italy, still ongoing in the Campanian region, is associated with melting of mantle domains that had been metasomatised by melts and fluids derived from sediments and/or subducted lithosphere[13,15–18]. The heat flux required to induce melting of the metasomatised wedge can be attributed to rising hot asthenosphere, promoted by formation of slab-tear faults, slab windows and detachment[19,20].

Despite extensive studies of Italy's highly variable potassium-rich volcanic products, the exact nature and origin of the mantle source contaminants remains uncertain[18]. Recent studies of olivine-hosted melt inclusions have established their strength in determining the marked local heterogeneity of Italian magma sources, including the detection of previously unidentified components[4,7,21,22]. Nikogosian and Van Bergen, 2010[23], revealed the presence of multiple, geochemically distinct, populations of melt inclusions in individual lava samples from Roccamonfina volcano and the Ernici volcanic field within the Roman Magmatic Province, the central part of the Italian subduction system. These volcanic centres, located above a slab-tear window[17,20] produced rock series originating from parental magmas with an extremely large range of K2O (0.5–10 wt.%)[15,24], originally grouped into silica-undersaturated high-potassium (HKS) and silica-saturated low-potassium (LKS) series[25]. Olivine-hosted melt inclusions from mafic HKS lavas, selected to exclude processes occurring in the crust, show strong compositional

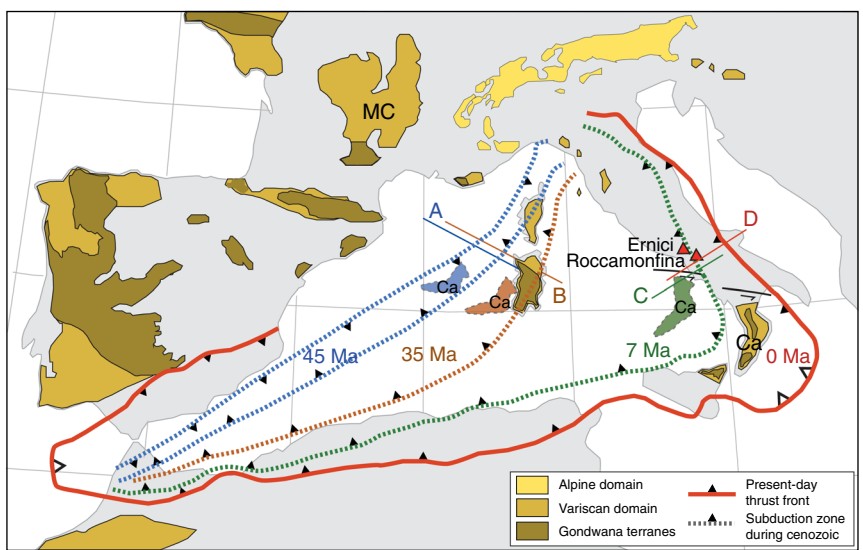

**Fig. 1** Schematic map of the Western Mediterranean indicating the Roccamonfina and Ernici volcanoes. The migration of the subduction zone since the Eocene (stippled lines) and the presence of Variscan and peri-Gondwanan terranes in the region are indicated (modified after refs. [8,10,14,64,75]). The arc changed subduction polarity from south-east to northwest at around 45 Ma (blue) and migrated eastward to its indicated position at 35 Ma (brown). Continued subduction and roll-back caused significant fore- and back-arc extension and migration of the Calabria segment (Ca) towards the East at 7 Ma (green), and to its current location to meet the Dinarides continental lithosphere at 0 Ma (red). Regions of the thrust front with active subduction are indicated with larger open symbols. Coloured lines with labels A–D represent the locations of profile cartoons in Fig. 5. Red triangles represent the current location of Roccamonfina and Ernici on peninsular Italy. MC: Massif Central

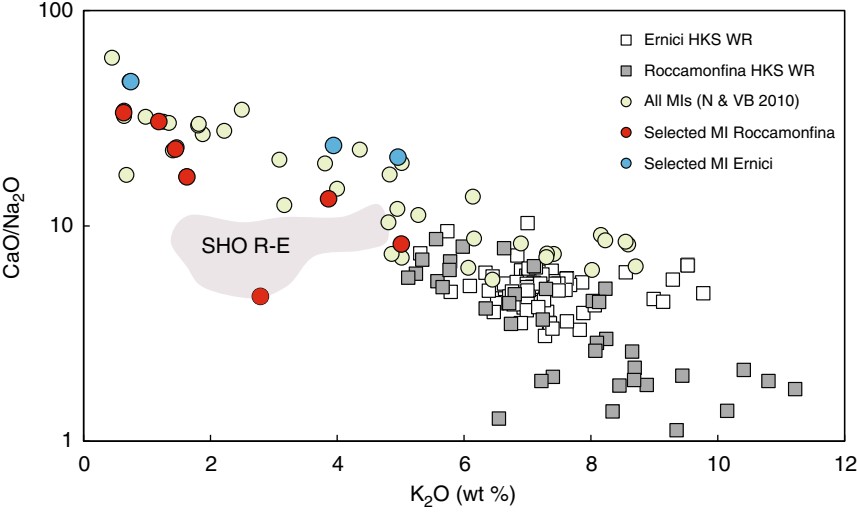

**Fig. 2** $K_2O$ versus $CaO/Na_2O$ in HKS lavas and melt inclusions from Roccamonfina-Ernici. A significant portion of the high Fo olivine-hosted melt inclusions suggests mixing with a component that has relatively high $CaO/Na_2O$ and low $K_2O$. The red symbols indicate the subset of the melt inclusions selected for Sr-Nd-Pb isotope analyses to investigate the exotic mantle component. The bulk rock lava data[76] and field for shoshonitic lavas at Roccamonfina-Ernici[76] (SHO R-E) in grey is shown for comparison. The MI source data[23] are provided as a Source Data file

diversity inferred to reflect mixing between melts derived from a mantle source with imprints of siliceous potassium-rich as well as calcium-rich potassium-poor subduction components[23]. The latter component was not recognised in bulk lava compositions but trace-element compositions suggest a carbonatite like affinity.

Here we present coupled Sr-Nd-Pb isotope data on individual olivine-hosted melt inclusions from the HKS lava series of Roccamonfina and Ernici, analysed by thermal ionisation mass spectrometry (TIMS) making use of $10^{13}$ Ω amplifier technology[7,26] and a Pb double spike technique[27]. The new analytical advances allow determination of isotopic heterogeneity of the primitive melts. These isotopic data are complimented by published major and trace-element data[23]. The MI data reveal a previously unknown exotic component with an extremely unradiogenic Pb isotope composition which we suggest is derived from ancient recycled lower continental crust (LCC). Our interpretation of possible sources and pathways for emplacement of this LCC component in the Italian mantle source are discussed within the framework of the geodynamic evolution of the western European and Mediterranean realm since the late Ordovician when micro-continental terranes were rifted from the northern margin of Gondwana and amalgamated onto southern Europe[11]. We suggest that isotope data on primitive olivine-hosted melt inclusions can provide crucial information to better understand subduction-related geodynamic processes that control fluxes to and from the mantle.

## Results

**Isotope compositions of melt inclusions.** Twenty melt inclusion-bearing olivine grains from Roccamonfina and Ernici HKS lavas were analysed for isotopic compositions. The samples were selected on the basis of MgO contents of both the host olivine ($Fo_{87}$–$Fo_{90}$) and melt inclusions (7.0–9.1 wt.% MgO) and to include a distinct group of inclusions with relatively low $K_2O$ and $Na_2O$ but high CaO and distinct trace-element contents (see Supplementary Data 1, Fig. 2 and[23]). The selection process aimed to specifically include early crystallised olivine to minimise potential effects of crustal assimilation processes. For full details of the MI populations of Roccamonfina and Ernici HKS lavas the reader is referred to the study by Nikogosian and Van Bergen[25].

Melt inclusion Sr-Nd-Pb isotope data are presented in Table 1 and Fig. 3. The $^{87}Sr/^{86}Sr$ ratios for the Roccamonfina inclusions range from 0.7096 to 0.7099 and for Ernici from 0.7075 to 0.7097. The $^{143}Nd/^{144}Nd$ ratios for Roccamonfina range from 0.51197 to 0.51218, and for Ernici from 0.51204 to 0.51226. Pb isotope ratios are highly variable: for Roccamonfina $^{206}Pb/^{204}Pb = 17.8$–18.4; $^{207}Pb/^{204}Pb = 15.5$–15.7 and $^{208}Pb/^{204}Pb = 37.5$–38.4 and for Ernici $^{206}Pb/^{204}Pb = 17.8$–18.8; $^{207}Pb/^{204}Pb = 15.6$–15.7 and $^{208}Pb/^{204}Pb = 37.7$–39.0. Sr and Nd isotope compositions of the selected inclusions largely overlap those of HKS lavas from the two volcanoes. Two MIs from Ernici, however, have lower Sr and slightly higher Nd isotope compositions, suggesting more shoshonitic characteristics. In contrast to the overall similarity of Sr and Nd isotope ratios between MIs and bulk lavas, the Pb isotope compositions of the inclusions are consistently more unradiogenic than the lavas. Ernici MIs record the widest range of compositions, from approaching the bulk lava to the most unradiogenic composition. Roccamonfina inclusions are all significantly more unradiogenic than the host lavas.

A sub-set of the MIs analysed for Sr-Nd-Pb were previously analysed for major and trace elements[23] (Supplementary Data 1). When combining the MI data from the two volcanic centres, and ignoring what might be an outlier (ERN4–345), $^{206}Pb/^{204}Pb$ ratios correlate inversely with the $CaO/Al_2O_3$ ratio and the contents and ratios of several trace elements (e.g., Sr, Th, U, REE, Sr/Nd, U/Ce) (Fig. 4). Furthermore, $^{143}Nd/^{144}Nd$ ratios of the MIs generally correlate inversely with Sr, REE, Th and U contents and the Sr/Nd ratio (Fig. 4 and Supplementary Data 1). There are no conspicuous relationships between $^{87}Sr/^{86}Sr$ and chemical signatures of the MIs.

## Discussion

Based on the characteristics of the host lavas, olivines and MIs[23] effects from shallow crustal contamination, as has been inferred for some of the Neogene-Quaternary volcanic centres in Italy[28–30], can be ruled out. There is no petrographic evidence for crustal assimilation; the host olivines have primitive compositions (Fo > 87 mol%) and the trace element and isotope characteristics of the MIs cannot be explained by mixing with Italian limestone in the crust (see Supplementary Note 1 and Supplementary Figs. 1–4 for

**Table 1 Sr-Nd-Pb isotope compositions of Roccamonfina–Ernici melt inclusions**

| Sample | $^{87}Sr/^{86}Sr$ | $2SE \times 10^{-6}$ | $^{143}Nd/^{144}Nd$ | $2SE \times 10^{-5}$ | $^{206}Pb/^{204}Pb$ | 2SE | $^{207}Pb/^{204}Pb$ | 2SE | $^{208}Pb/^{204}Pb$ | 2SE |
|---|---|---|---|---|---|---|---|---|---|---|
| R7–130 | 0.709829 | 8 | 0.51206 | 4 | 18.401 | 0.028 | 15.588 | 0.033 | 38.428 | 0.103 |
| R7 134 | 0.709783 | 9 | 0.51212 | 6 | 18.050 | 0.003 | 15.631 | 0.002 | 38.104 | 0.007 |
| R7–135 | 0.709874 | 9 | 0.51211 | 3 | 18.071 | 0.004 | 15.634 | 0.003 | 38.148 | 0.010 |
| R7–136 | 0.709794 | 10 | 0.51218 | 4 | 18.163 | 0.015 | 15.612 | 0.015 | 38.197 | 0.045 |
| R7–139 | 0.709624 | 37 | 0.51197 | 10 | 18.170 | 0.013 | 15.621 | 0.013 | 38.250 | 0.038 |
| R7–152 | 0.709802 | 9 | 0.51210 | 3 | 18.293 | 0.006 | 15.650 | 0.006 | 38.421 | 0.016 |
| R7–184 | 0.709860 | 9 | 0.51209 | 5 | 18.297 | 0.032 | 15.632 | 0.028 | 38.411 | 0.076 |
| R7–267 | 0.709850 | 34 | 0.51210 | 29 | 17.745 | 0.090 | 15.500 | 0.075 | 37.514 | 0.214 |
| ERN5–135 | 0.709662 | 21 | 0.51212 | 13 | 17.791 | 0.019 | 15.565 | 0.018 | 37.714 | 0.050 |
| ERN5–145 | 0.709746 | 9 | 0.51211 | 6 | 18.751 | 0.004 | 15.695 | 0.004 | 38.997 | 0.012 |
| ERN4–31 | 0.709587 | 10 | 0.51210 | 10 | 18.677 | 0.023 | 15.715 | 0.026 | 38.955 | 0.078 |
| ERN4–46 | 0.709728 | 10 | 0.51209 | 7 | 18.589 | 0.025 | 15.704 | 0.031 | 38.836 | 0.100 |
| ERN4–68 | 0.709740 | 14 | 0.51204 | 8 | 17.999 | 0.004 | 15.620 | 0.004 | 38.058 | 0.013 |
| ERN4–85 | 0.709676 | 8 | 0.51212 | 8 | 18.459 | 0.007 | 15.669 | 0.009 | 38.632 | 0.028 |
| ERN4–345 | 0.708580 | 21 | 0.51226 | 11 | 18.037 | 0.006 | 15.611 | 0.006 | 38.044 | 0.019 |
| ERN4 352 | 0.709671 | 11 | 0.51206 | 5 | 18.076 | 0.021 | 15.655 | 0.019 | 38.246 | 0.053 |
| ERN4–354 | 0.709726 | 8 | 0.51214 | 3 | 18.585 | 0.006 | 15.676 | 0.006 | 38.758 | 0.016 |
| ERN4–358 | 0.707485 | 15 | 0.51221 | 10 | 18.465 | 0.009 | 15.660 | 0.008 | 38.603 | 0.023 |
| ERN4–362 | 0.709770 | 12 | 0.51211 | 11 | | | | | | |
| ERN4–363/351[a] | 0.709729 | 6 | 0.51215 | 3 | 18.708 | 0.009 | 15.678 | 0.008 | 38.905 | 0.024 |
| AGV1–1[b] | 0.703989 | 14 | 0.51279 | 5 | 18.942 | 0.005 | 15.664 | 0.004 | 38.585 | 0.011 |
| AGV1–2[b] | 0.704020 | 13 | 0.51277 | 6 | 18.942 | 0.005 | 15.661 | 0.004 | 38.572 | 0.012 |

[a]This sample included two melt inclusions with similar characteristics to have enough material for isotope analyses
[b]The AGV1 samples were processed as aliquots from a single dissolution to contain 500 pg Pb, 400 pg Nd and 8 ng Sr

a detailed discussion). We thus interpret the combined Sr-Nd-Pb data in MIs from Roccamonfina-Ernici olivines to record the presence of heterogeneity in the mantle source and infer melt supply from distinct mantle components.

The Sr and Nd isotope compositions of the HKS melt inclusions closely resemble those of HKS bulk lavas from Roccamonfina-Ernici and other volcanoes in the Roman Magmatic Province (RMP). Collectively, these RMP rocks have intermediate compositions within the spectrum of Italian Neogene-Quaternary volcanics that display a general trend from high $^{87}Sr/^{86}Sr$ and low $^{143}Nd/^{144}Nd$ ratios in the North (Tuscan Magmatic Province) to low $^{87}Sr/^{86}Sr$, high $^{143}Nd/^{144}Nd$ ratios in the South (Etna; Aeolian Islands) (Fig. 3d). This regional isotopic variability, based on bulk rock compositions alone, is generally interpreted as a mixing array between a HIMU-like mantle endmember and upper continental crust material[31]. All HKS rocks show strong fractionation between LILE and HFSE typical for melts derived from a mantle domain that experienced addition of subducted sediment components[32,33]. The ultra-potassic signature in combination with silica-undersaturation and calcium enrichment is attributable to a mixed pelitic-carbonate nature of sediments that delivered the metasomatising agents[34]. The significant proportion of carbonates in subducted lithologies is also expressed in their contribution to the ongoing strong geological $CO_2$ emission that marks the entire Tyrrhenian border of Central-Southern Italy[24,35,36].

In contrast to the Sr and Nd isotope data, the Pb isotope compositions of the primitive melt inclusions deviate strongly from the regional mixing array and record a far larger range than the lavas. In diagrams of $^{207}Pb/^{204}Pb-^{206}Pb/^{204}Pb$, $^{87}Sr/^{86}Sr-^{206}Pb/^{204}Pb$ and $^{143}Nd/^{144}Nd-^{206}Pb/^{204}Pb$ (Fig. 3a–c) this isotopic variability can be explained by binary mixing between melts from the sediment-metasomatised mantle source, dominant for HKS magma in the RMP, and an exotic source component, not readily recognised in data from bulk lava samples. The observed isotopic variability testifies that accumulation and efficient mixing of fractional melts before eruption effectively masks the full heterogeneity of the mantle source in data from bulk lava samples. Relations between the isotope ratios of the MIs and their incompatible element signatures (Fig. 4 and Supplementary Data 1) suggest that source heterogeneity at Roccamonfina-Ernici controls both isotopic composition and elemental concentration. Based on the combined geochemical variations in the measured MIs, the exotic source component with highly unradiogenic Pb also has high Sr, Th, U, REE contents and $CaO/Al_2O_3$, Sr/Nd, Th/Pb, U/Ce and La/Yb ratios, relative to the endmember that sourced the bulk of the HKS magma. For incompatible elements, however, concentrations and inter-element variations in the extracted melts are also a function of the degree of partial melting that is controlled by the melt temperature and melt productivity of the source mineralogy[37,38]. If the source components have different mineral assemblages they will be characterised by different melting behavior. Variation in source mineralogy in the mantle below Ernici and Roccamonfina is expected since the subduction-related metasomatic imprints are likely not pervasive but expressed in a network of veins with mineralogies and melting temperatures different to those of surrounding wall rock[15,23,39]. For such a veined system, it is conceivable that early melt fractions were extracted from mineralogically and geochemically distinct, more fusible, lithological domains present in the source of HKS magmas. Hence, the varying trace-element signatures of the MIs are likely inherited both from compositional and mineralogical differences between the source components and from entrapment of the mixed melts during increasing degrees of partial melting[40]. The apparent relations between trace-element ratios (e.g., La/Yb and Sr/Nd) and Pb isotope ratios (Fig. 4) suggest such a coupling between the degree of melting and sampling of the exotic source component. This component contributed to the entrapped primary melt at small degrees of partial melting but appears to have become progressively diluted during continued melting of the mantle domain. Bulk lavas tend to plot off the mixing trends defined by the MIs (Fig. 4) due to the role of intra-crustal differentiation and mixing processes, in line with their less magnesian and generally more evolved compositions.

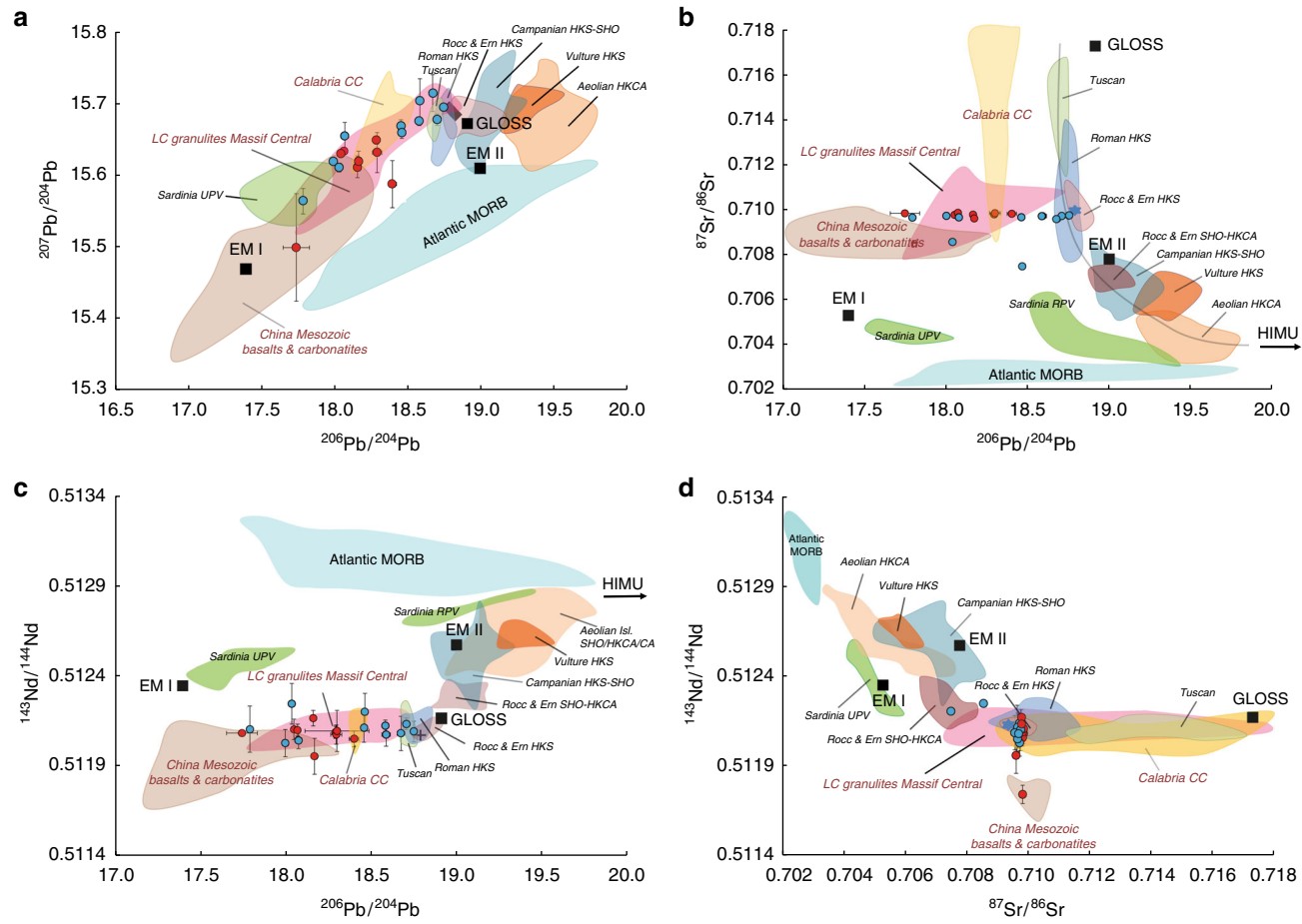

**Fig. 3** Sr-Nd-Pb isotope data for the analysed melt inclusions. **a** $^{207}Pb/^{204}Pb$ versus $^{206}Pb/^{204}Pb$; **b** $^{87}Sr/^{86}Sr$ versus $^{206}Pb/^{204}Pb$; **c** $^{143}Nd/^{144}Nd$ versus $^{206}Pb/^{204}Pb$; **d** $^{143}Nd/^{144}Nd$ versus $^{87}Sr/^{86}Sr$. Error bars represent 2SE. Data are compared to fields for Italian lavas[76,77], Calabrian Continental Crust[78], Atlantic MORB[79], Massif Central Lower Crustal granulites[80,81], Mesozoic Chinese basalts and carbonatites[49,82,83], EM-1 and EMII[84], and GLOSS[33]. MI source data are provided as a Source Data file

Lead mass balance considerations establish that the exotic source component contributes only a minor fraction of melt (<5%) and consequently is not readily identified from bulk lava compositions. In addition, the relatively small spread in Sr isotope ratios within the mixing array, together with the negative correlation between $^{206}Pb/^{204}Pb$ ratios and Sr concentrations (Fig. 4f) suggests high Sr contents in the exotic source and appears to rule out a major difference in Sr isotope compositions between the two end-members. Binary mixing relations between endmember melts using the concentrations of our most extreme melt inclusions (R-139 with unradiogenic $^{206}Pb/^{204}Pb$, Sr = 2820 ppm; Pb = 15 ppm and E-363 with $^{206}Pb/^{204}Pb$ similar to the host lava, Sr = 349 ppm; Pb = 15 ppm) can only explain the relatively constant Sr isotope compositions of our MI if the two endmember melts have a similar Sr isotope composition. We thus infer that the exotic mantle component has a Sr isotope composition comparable to that of the bulk lavas of Roccamonfina and Ernici. Its Nd isotope composition is difficult to assess in absence of an obvious relationship with the Pb isotopes in the MIs (Fig. 3c). However, the $^{143}Nd/^{144}Nd$ is likely lower than that of the dominant source component since some MIs have significantly lower $^{143}Nd/^{144}Nd$ than the bulk lavas (Fig. 3d)[7]. Furthermore, there is an inverse correlation between trace-element concentrations and Nd isotopes, similar to that expressed by the Pb isotopes (Fig. 4). From these observations we conclude that primary HKS magma of Roccamonfina and Ernici originated from a mantle domain that not only carries an imprint from subducted pelitic-carbonate

sediment but also hosts a minor component with a distinct unradiogenic Pb and Nd isotope composition.

The unradiogenic Pb isotope composition that marks the exotic component suggest derivation from material that had lost U with respect to Pb in an ancient event, thereby freezing-in the Pb isotope composition at the time of the event. Uranium loss commonly occurs during high-grade metamorphic processes within lower continental crust (LCC) during orogenic events[41,42] making ancient LCC a viable source for the unradiogenic Pb. An alternative scenario proposed to explain unradiogenic Pb in the mantle source of some ocean island basalts (EM-1 type isotopic composition), is that ancient pelagic sediments experienced processing by preferential U dehydration during a >2Ga subduction event[43]. This sedimentary component, which develops its unradiogenic character over time, is later returned to the surface by a deep seated mantle plume[44,45]. We consider this mechanism highly unlikely for the studied area, given the lack of geochemical[31] and geophysical[14] evidence for plume activity below central Italy.

The involvement of modern subducted sediments in the magma source region is, however, to be expected. If these modern sediments would be sourced from ancient Archaean crust with unradiogenic Pb due to low time-integrated U/Pb and Th/Pb ratios[46] their subduction could represent an alternative unradiogenic Pb source to LCC. However, because the nearest Archaean craton lies > 2000 km away from Italy, these type of Archaean sediments are absent in the Mediterranean realm. Furthermore, such Precambrian sediments would be characterised by extremely low Nd isotope compositions

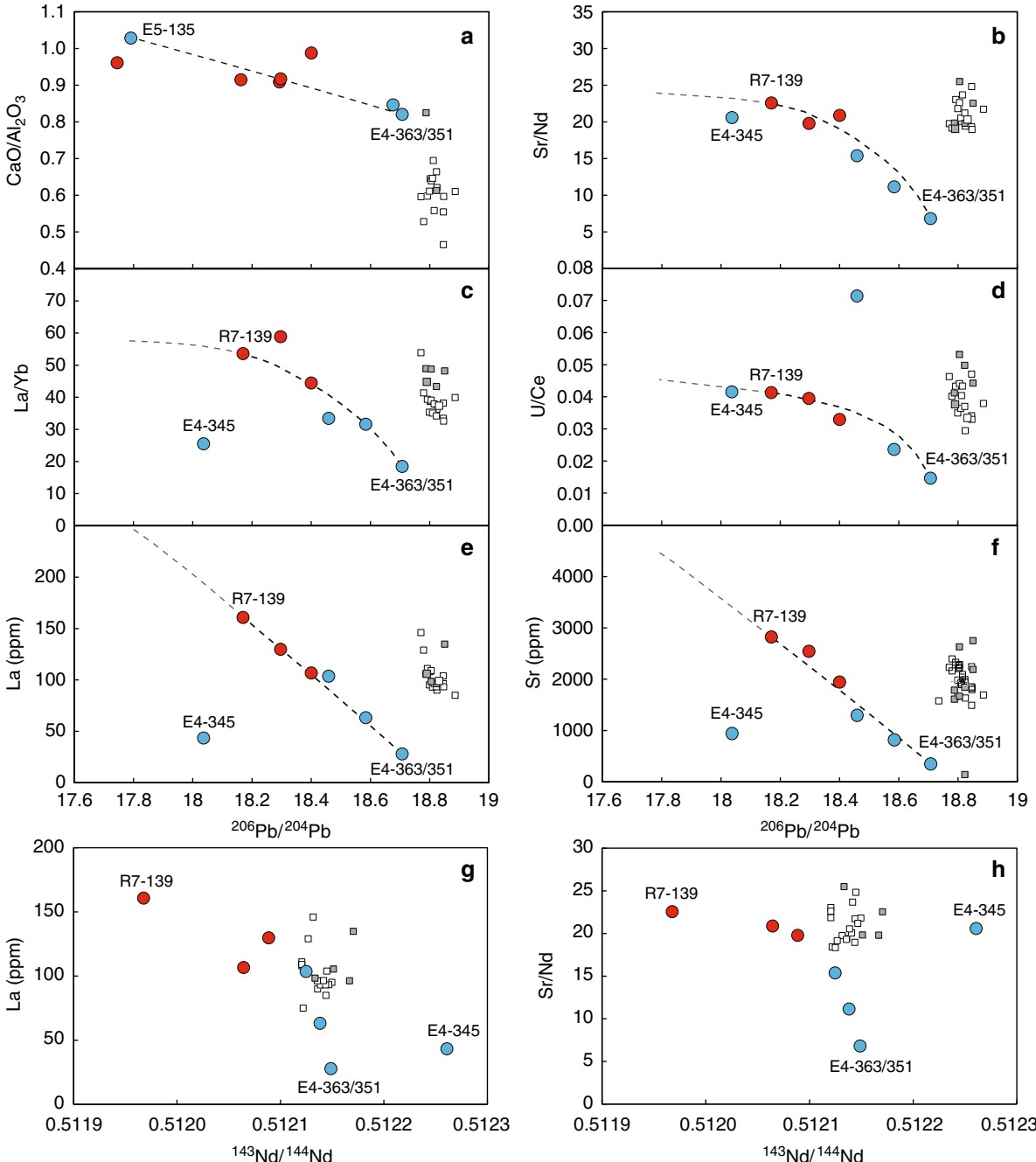

**Fig. 4** Relations between $^{206}Pb/^{204}Pb$ or $^{143}Nd/^{144}Nd$ and major and trace-element contents and ratios for MIs compared to Roccamonfina Ernici bulk rock lavas. The upper six panels **a**–**f** show $CaO/Al_2O_3$, Sr/Nd, La/Yb, U/Ce, La (ppm) and Sr (ppm) versus $^{206}Pb/^{204}Pb$; the lower two panels **g**, **h** show La (ppm) and Sr/Nd versus $^{143}Nd/^{144}Nd$. Grey and white square symbols represent bulk rock data from Roccamonfina and Ernici, respectively[76]. Red and blue circles are melt inclusion data for Roccamonfina and Ernici from this study. Negative relations exist between Pb isotopes and the element concentrations and ratios, suggesting mixing between a source with similar to but more trace element depleted compositions than the host lavas, and a source with unradiogenic Pb and highly enriched trace-element contents. Dashed lines represent binary mixing between the melt inclusion with most radiogenic $^{206}Pb/^{204}Pb$ (E4-363-351) and the inclusion along the trend with most unradiogenic $^{206}Pb/^{204}Pb$ (E5-135 or R7-139, ignoring the outlier E4-345). The lines are extended towards the potential exotic endmember with more unradiogenic Pb (grey dashed line). Source data are provided as a Source Data file

($εNd < −20$)[46] unlike the compositions of our melt inclusions ($εNd ∼ −10$). We thus infer that recycling of Paleozoic LCC material into the mantle, either as a fragment of lithosphere or in the form of sedimentary erosion products following an exhumation event, is the most plausible mechanism for creating the observed Pb isotope signature[47–49]. Negative correlations between $^{206}Pb/^{204}Pb$ and $CaO/Al_2O_3$ and trace elements such as Sr, U, REE of Roccamonfina-Ernici MIs (Fig. 4; Supplementary Data 1) suggest

that carbonate-rich material, presumably as sediments, was added to the unradiogenic LCC component after the ancient orogenic event.

The only volcanic products, i.e. lavas, within the Mediterranean realm that have unradiogenic Pb isotope compositions similar to the Roccamonfina-Ernici melt inclusions are Late-Cenozoic lavas from Sardinia (UPV group)[50]. Here, melts derived from delaminated and detached ancient LCC are thought to have interacted with peridotite to produce an orthopyroxene-rich mantle rock

that was the source of the UPV lavas[51]. Apart from the comparable unradiogenic Pb isotope signatures, the Roccamonfina-Ernici endmember is, however, geochemically distinct to the Sardinian lavas, being characterised by more unradiogenic Nd and radiogenic Sr isotope ratios (Fig. 3) as well as lower $SiO_2$, higher CaO (and $CaO/Al_2O_3$), lower Ni and HFSE, and higher REE, La/Nb and Th/Nb (Supplementary Data 1).

Supporting evidence for involvement of LCC components in the sources of Italian K-rich magmas comes from MIs in olivine from Latera volcano in the northern Roman Magmatic Province[4]. Relative to bulk lavas from any of the Neogene-Quaternary volcanic centres in Italy, the MIs have exceptionally variable Pb isotope compositions with two unradiogenic $^{206}Pb$ populations at different levels of $^{207}Pb/^{204}Pb$. The MI compositions suggest mixing with two different unradiogenic Pb components; one with moderately radiogenic $^{207}Pb$ and $^{206}Pb$ similar to lower continental crust found in the Variscan and older basement of Sardinia and Calabria, and another with extremely unradiogenic $^{207}Pb/^{204}Pb$ values is requiring an older LCC component, presumably of Precambrian age. Compared to Latera our MIs agree with the data that have moderately radiogenic $^{207}Pb$ and $^{206}Pb$, but reach more extreme unradiogenic $^{206}Pb/^{204}Pb$ values. The Latera MI compositions are, similar to Roccamonfina-Ernici, not recorded by bulk samples of host lava.

Outside the Mediterranean realm, Mesozoic mantle-derived mafic rocks from the North China Craton[49,52] represent a rare case of magmatic rocks with unradiogenic Pb isotope compositions and Sr and Nd isotope ratios comparable to the melt inclusions studied here (Fig. 3). The isotopic signatures of these magmas have been explained by metasomatic modification of their mantle source by addition of melts from LCC material, introduced either by subduction or delamination[53,54].

To explore the provenance of the inferred LCC component in a geodynamic context and thus better understand subduction recycling processes, we evaluate the potential mechanisms of emplacement of LCC within the mantle source below Italy. High-grade metamorphic crust was formed in the region during the Variscan Orogeny that resulted from collision of super terranes Gondwana and Laurussia in Permian times and in preceding orogenic events on the northern margin of Gondwana in the Ordovician[8–12]. Multiple continental blocks with remnants of these old orogenic events form part of the modern central-southern European continent (e.g., Massif Central, Sardinia, Corsica, Calabria, Adria, Fig. 1). The Sr-Nd-Pb isotope signatures of our melt inclusions overlap in part with composition of granulitic lower crust as recorded by xenoliths from the Massif Central[55,56] (Fig. 3), one of the massifs representing the exhumed internal zone of the Variscan Orogeny (Fig. 1). We infer that recycled lower crustal material, with isotopic characteristics similar to that of the Massif Central, but more unradiogenic in terms of Pb and Nd isotopes, must be present in the mantle source that produced the Roccamonfina-Ernici magmas. Below we discuss several possible mechanisms and pathways within the framework of the Cenozoic geodynamic evolution of the Western Mediterranean that could have been responsible for transporting a Variscan or peri-Gondwana LCC component to the mantle wedge beneath peninsular Italy.

Delamination (or subduction erosion) of a LCC segment from an ancient continental block after the Variscan Orogeny[57] could have resulted in lithosphere metasomatism by melts extracted from the transported segment, similar to the process inferred for the unradiogenic-Pb volcanics of Sardinia[47]. The lack of a deep crustal root beneath the Massif Central (crustal thickness 25–30 km)[58] may reflect thermal relaxation and/or delamination of lower crust and mantle lithosphere following the orogenic phase[59]. European lithosphere, metasomatised in this way, may

have been involved in the early southeastward Cretaceous-Oligocene Alpine subduction (scenario 1 at ~45 Ma Fig. 5a)[13,60]. Slab roll-back in the Apennine subduction system, following cessation of Alpine subduction and polarity change, should then have moved the LCC metasomatic material beneath the Italian Peninsula. Alternatively, contamination of the mantle through direct subduction of LCC could also have occurred during the subduction of "European" continental lithosphere under northern Italy as has been inferred from geological reconstructions[60] (scenario 2, Fig. 5a).

In a third scenario, uplift of Paleozoic basement blocks within the old Alpine, Apennine and/or Dinaride belts provided suitable conditions for erosion of exhumed LCC lithologies, as well as a favourable paleogeography for transport of the resulting unradiogenic sediments into the early Adriatic Sea[10,61,62]. Westward subduction in the Apennine system and subsequent mobilisation of the LCC-derived sediments could have metasomatised the overlying lithospheric mantle transferring the unradiogenic Pb signature (~35 Ma, Fig. 5b). This recycling route would be similar but older compared to the more recent metasomatism by the upper crustal sediment mixture that is thought to be responsible for the imprint in the RMP mantle source. Admixture of Adriatic marine carbonates to the LCC sedimentary subduction component could explain the virtually constant Sr isotope compositions of the MIs comparable to marine carbonates. At the same time, it would explain the apparent enrichment of the exotic component with elements such as Ca, Sr, U.

Lower continental crust material could also have been derived from the old Calabrian crustal segment with Variscan and Gondwanan basement relics[63] that is currently located ~300 km south-east of Roccamonfina-Ernici. Kinematic reconstructions of the Western Mediterranean indicate that the Calabrian segment, initially neighbouring Sardinia, moved to its current position through roll-back during the Miocene (Fig. 1)[13,64]. It is conceivable that, in the course of the roll-back history, thermo-mechanical erosion of Calabrian continental lithosphere, induced by toroidal mantle flow around a slab-tear fault[65–67] transferred isolated domains of LCC material to the magma source in the mantle wedge (scenario 4, Fig. 5c). Currently exposed rocks in Calabria show significant variability in Sr and Nd isotope compositions that overlap with Roccamonfina-Ernici lavas and inclusions, but their Pb isotope compositions are more restricted and less unradiogenic than the most extreme MIs (Fig. 3).

Finally, LCC material from the Adriatic margin could have been introduced into the mantle during the final stages of the Apennine convergence and collision over the last 5 Ma. Subduction of Adriatic continental margin and delamination within the downgoing plate by physical decoupling from the upper crust may have exposed slivers of LCC to hot asthenosphere, as has been invoked for the Neogene-Quaternary magmatism of the northern Apenninic arc[68]. Opening of a slab window or slab tearing beneath the Roman province[14,19,69] and subsequent inflow of hot asthenosphere may have facilitated mobilisation and transfer of subducted continental material into the Roccamonfina-Ernici mantle source (scenario 5, Fig. 5d) as predicted by thermo-mechanical evolution models[70–72]. Whether the age and composition of Adriatic lower crust in the Roccamonfina-Ernici sector complies with the isotopic signatures of the exotic component remains to be assessed. Although Variscan basement is known to occur below Sardinia and in the Tuscan region of the northern Apennines[10] (Fig. 1), there is no evidence for its presence in the Roccamonfina-Ernici area.

Although the specific mechanism responsible for adding an ancient LCC component to the mantle wedge beneath the Italian Peninsula cannot unambiguously be constrained, these scenarios illustrate the multitude of potential pathways for the transfer of Variscan or older material to mantle domains at depth where it

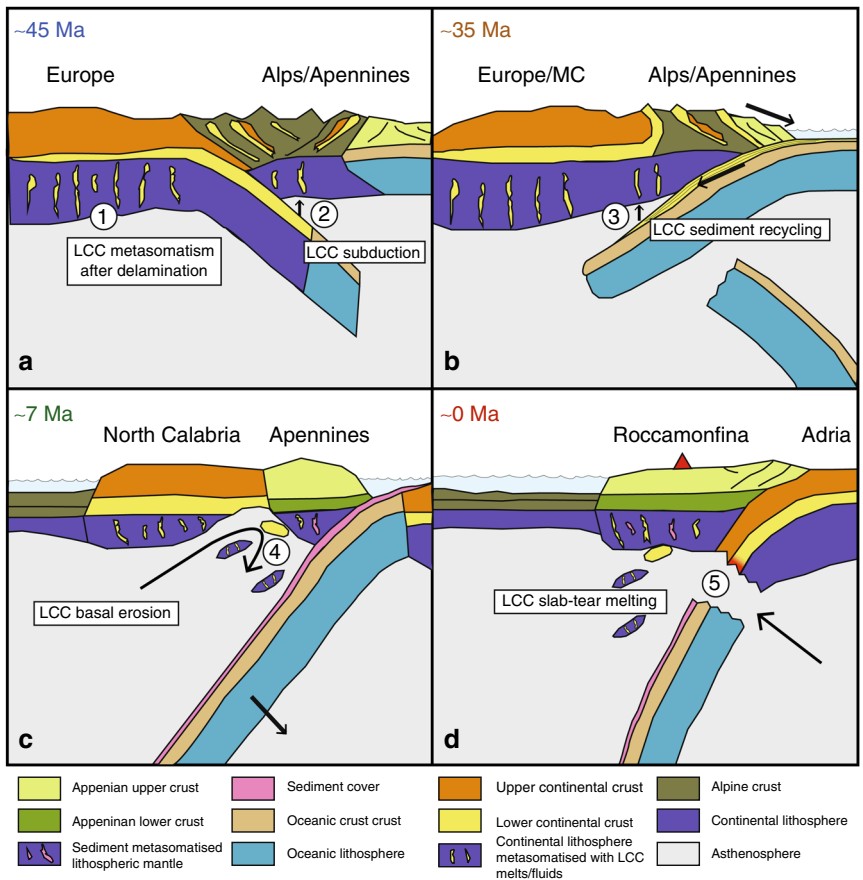

**Fig. 5** Provenance of the lower continental crust (LCC) in a geodynamic context. Schematic cross-sections illustrate potential geodynamic processes that resulted in the emplacement of a LCC component that we infer to be present in the Roccamonfina-Ernicic melt source. **a** Lithosphere and LCC delamination after the Variscan Orogeny below Europe and the Massif Central (MC) led to metasomatism with a LCC imprint (scenario 1). Alternatively, European LCC subducted during the onset of the Alpine collision before subduction reversal between 45 and 35 Ma (scenario 2). **b** Subduction recycling of sediments from exhumed lower crustal blocks in the hinterland during and after the Alpine Orogeny (scenario 3). **c** Accelerated roll-back along a transform or slab-tear fault results in thermo-mechanical erosion of the base of Calabria between 8–6 Ma (scenario 4). **d** Slab tear and inflow of hot asthenosphere results in melting of Adriatic LCC (scenario 5). Note that although Variscan LCC is known to occur below Sardinia and in the Tuscan region of the northern Apennines[10] (Fig. 1), there is no evidence for its presence in the Roccamonfina–Ernici area

would partially melt to form the exotic component recorded in the MI population.

The Sr-Nd-Pb isotope data in high Fo-olivine hosted individual melt inclusions from two adjacent Quaternary volcanic centres in mainland Italy reveal much larger compositional variability than their whole rock lavas. We suggest that the relations between Sr-Nd-Pb isotopes and major and trace-element concentrations are a result of binary mixing between primary melts from a sediment-metasomatised mantle component and a previously unrecognised exotic component, not detected in bulk lavas. The exotic component has extremely unradiogenic Pb, and low alkali contents relative to the voluminous high-potassium magmas of central-southern Italy, and is interpreted to be derived from recycled lower continental crust material that had lost U in an ancient event. We infer that a lower continental crustal unit, formed during the Variscan Orogeny, or earlier, has been recycled into the recent melt source beneath the Italian Peninsula. The Cenozoic subduction and collision history of the western Mediterranean with its long dominance of continental convergence potentially resulted in the introduction of LCC material into the upper mantle in multiple ways. Transport, mobilisation and infiltration of LCC-derived components into current magma sources could have been promoted by subduction of continental lithosphere or its sedimentary erosion products, subduction erosion, or slab-detachment. The Sr-Nd isotope signatures, similar or close to that of HKS magmas in the Roman Magmatic,

were either a property of the original LCC lithology or were admixed in a marine environment if the LCC material was subducted in the form of sediment eroded from an exhumed crustal domain. Our results demonstrate that isotope data on individual melt inclusions in primitive high forsterite olivine can provide insight into tectono-magmatic processes that govern fluxes in and out of the mantle.

## Methods

**Sample preparation.** All host olivine and exposed homogenised MI's were analysed by EMP to determine major element compositions. A subset of inclusions were analysed by SIMS and/or LA-ICPMS for trace-element contents[23]. The melt inclusions were selected to contain a minimum of 30 pg Nd, 2 ng Sr and 100 pg Pb to minimize a potential blank effect. For isotopic analyses the olivine grains free of adhering groundmass, were taken out of epoxy holders and olivine and melt inclusions were dissolved in Teflon vials in a mix of HF and $HNO_3$. Before miniaturised Sr-Nd isotope separation[7] the sample Pb fraction was separated using 50 µL Biorad AG1-X8 200–400 mesh resin following the methods of[27]. Due to the small sample size (<5 mg) and to minimise the blanks a single column pass was used. Total procedural blanks processed during this study gave averages of 10.4 pg for Sr ($n = 2$); 0.11 pg for Nd ($n = 2$) and 1.75 pg for Pb ($n = 4$).

**Analytical techniques.** Sr-Nd-Pb isotope analyses were performed on a Thermo Scientific Triton *Plus* equipped with four $10^{13}$ Ω amplifiers[7,26,73]. Pb analyses were performed using the double spike techniques as described in ref. [27], samples were run to exhaustion. Two aliquots of international rock standard AGV-1 taken from a single dissolution were processed along with the melt inclusions. The aliquots that were loaded on the columns contained 500 pg Pb, 400 pg Nd and 8 ng Sr and

were similar in size compared to the melt inclusions. The Sr-Nd-Pb data obtained for AGV-1 (Table 1) is in excellent agreement with the preferred values on the GeoREM database website[74].

## Data availability

All data presented in this manuscript are available in Table 1 and in the Supplementary Data 1.

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

## Acknowledgements

This project has received funding from the European Research Council (ERC) under the European Union's Horizon 2020 research and innovation programme (grant agreement No 759563 (ERC-StG ReVolusions) and 654208 (Europlanet 2020 RI)). We also ack-owledge financial support from the Netherlands Research Centre for Integrated Solid Earth Sciences (ISES) through grant 6.2.12 to IN and MJvB.

## Author contributions

J.M.K. performed the majority of the lab work and wrote the first version of the paper. I.N. performed experimental and analytical work (major-trace) to select the inclusions for isotope analyses. M.J.v.B. and I.N. collected the samples in the field. J.M.K., I.N., M.J.v.B., P.Z.V., and G.R.D. were all involved in data discussion and interpretation and edited the manuscript.

## Additional information

**Competing interests:** The authors declare no competing interests.

