## [Peer Review File · Nature Communications]

Reviewers' comments:

Reviewer #1 (Remarks to the Author):

This paper presents new isotope data for melt inclusions in olivine crystals from one of the Italian magmatic provinces, which have been of great interest to geochemists for a long time, in part because of their extreme isotopic and chemical compositions.

The paper focuses on two young volcanic regions, namely Roccamonfina and Ernici volcanic provinces, which are part of the Roman Magmatic Province.

The paper introduces a major analytical advance to the study of melt inclusions, namely a full set of isotope data for the elements Pb, Sr, and Nd. Previous authors were unable to measure full $^{206}\text{Pb}/^{204}\text{Pb}$, $^{207}\text{Pb}/^{204}\text{Pb}$ and $^{208}\text{Pb}/^{204}\text{Pb}$ ratios because their instrumentation lacked the sensitivity to measure the abundance of the ^{204}Pb peak with sufficient precision. I think this advance alone justifies a publication in Nature Communications. Also, the paper is well written and should be of interest to much of the isotope geochemistry community.

The interpretation of the data tends to follow previously established lines of reasoning, and it confirms and strengthens the evidence for an “exotic” source component tentatively identified in a previous paper by Nikogosian et al (2010). I generally do not try to argue about interpretations in a review of a paper, because I feel the authors are entitled to their own opinion, and my interpretive views may be wrong even though I am convinced I have the right answer. I will, in this case, make a partial exception to this general rule for myself, and I would like to encourage the authors to consider my arguments in a revised manuscript.

The authors make a case for the presence of ancient, recycled lower continental crust (LCC) in the source of the magmas, a component that is clearly identified only in the melt inclusions but is “buried” and not detectable in the whole rock compositions. I would agree that this is one possible model, which has in the past also been advocated for several so-called EM1-type oceanic basalts, with which the present rocks share several important characteristics. There is, however, an alternative explanation for the origin of at least some of these EM1-type basalts, which has also been explored extensively in the literature. This alternative is that EM1 isotopic characteristics (low $^{206}\text{Pb}/^{204}\text{Pb}$, high $^{208}\text{Pb}/^{206}\text{Pb}$, low $^{143}\text{Nd}/^{144}\text{Nd}$, intermediate $^{87}\text{Sr}/^{86}\text{Sr}$ ratios) can be explained by simple subduction and recycling of ancient pelagic sediments (e.g. Eisele et al. 2002, EPSL). This was always hard to distinguish from the alternative model invoking recycled lower continental crust, simply because the radiogenic isotopes allow both interpretations. For Pitcairn, however, the issue has been largely resolved by a recent paper by Delavault et al. (PNAS 113, 12952, 2016) who showed evidence for sulfur isotopes affected by mass-independent fractionation, a

phenomenon that is restricted to surface rocks older than about 2.4 Ga). Thus, it is pretty clear now that Pitcairn's EM1 signature is caused by recycled sediments, but this conclusion need not necessarily apply to all EM1 type basalts.

Nevertheless, with the "Pitcairn experience" as background, I ask myself, which property of the present set of data requires them to have a lower crustal source rather than an ancient sedimentary source. It seems clear from the high CaO/Al₂O₃ ratios (Nikogosian, 2010) that this source component must be carbonate rich – not really a characteristic of the lower continental crust, but a perfectly reasonable component of a subducted sediment. The component is particularly unradiogenic in Pb, a property that could characterize either a LCC component or an ancient sedimentary source. It is low in potassium, which might well be an indication of a sediment-derived primary carbonatite liquid. It has low Nb/U and Nb/U ratios, a property that might fit either source model. It is NOT particularly depleted in U or Th relative to other incompatible trace elements. It appears to have very slight negative Eu anomalies (Nikogosian, 2010), a property more in line with sediments than with lower crustal rocks. It does have positive Sr anomalies, which would point toward a gabbroic and therefore lower crustal source, but which might also be generated by the carbonatitic nature of the primary liquids. In summary, I find the evidence for a lower crustal source unconvincing, and I suggest that the sedimentary alternative is at least worth some discussion.

The authors allow for recycled sediments in one part of the discussion, but they immediately specify that these sediments must be derived from a lower crustal source.

I would insist that the source of such sediments must be old, but not necessarily derived from the lower crust. It seems to me that it would be hard to derive sediments directly from the lower crust. If the authors mean to derive these sediments from uplifted and eroded, originally lower crust, then this material has become part of the upper crust prior to erosion. Depending on the age of such crust, the Pb isotopes of sediments derived from it can be rather unradiogenic. For examples, the paper by Hemming et al. (2001, EPSL 184, 489-503) gives Pb isotopes for modern deep sea turbidites. Some of these turbidites have ²⁰⁶Pb/²⁰⁴Pb ratios as low as 16.8, although the vast majority of them do have ratios of 18 or greater. My point is that, if the sediments sample old rocks, they are likely to be rather unradiogenic in Pb. They do NOT have to come from the lower crust; they just need to be old.

There is another aspect of the manuscript that I find myself disagreeing with, namely the insistence that the recycled component, whatever it may be, has "metasomatized" the lithosphere, and it is ultimately the metasomatized lithosphere that melts. This view is extremely widespread among geochemists, but I find it is supported by remarkably little actual evidence. I don't think we know how to distinguish melts that ascend from the subducted slab directly to the surface from those that might first metasomatize the overlying lithosphere, which is then reactivated by some later time. And certainly, the tectonic situation in the Mediterranean region is extremely complicated, so I would find it hard to make a convincing case favoring one scenario over the other. Frankly, I think the model of "metasomatized lithosphere" has simply hardened into truth by frequent repetition in the literature rather than actual evidence. I am not going to insist on any reinterpretation of this aspect, but I do wish that the notion of the metasomatized lithosphere were not simply accepted as established truth.

A final point of this review concerns the treatment of the chemical composition of these basalts and their relationship to the newly measured isotopic compositions. I am writing this review from a place where I do not have access to the electronic supplement of Nikogosian's (2010) paper dealing with the chemistry of these melt inclusions, so I cannot make my own comparisons at the moment. I would consider it to be very helpful to the reader if the chemical data for those melt inclusions for which the authors have both isotopic and chemical data were listed again in an appropriate supplement, so that the reader who may want to look for alternative relationships does not have to dig through the literature and assemble a new Excel sheet to make plots. Specifically, I don't find the plots shown on Fig. 4 to be particularly instructive, because the absolute values of trace element concentrations can be affected by differences in the degree of melting, and I would hope to find better and more instructive correlations by plotting parameters such as CaO/Al₂O₃, Th/U, K/U, Nb/Th, Ce/Pb, Sr/Nd, Eu/Eu* versus ²⁰⁶Pb/²⁰⁴Pb ratios in order to specify the composition and nature of the exotic crustal component. For example, the linear variation of the isotopic compositions, which is best displayed by a diagram of ²⁰⁸Pb/²⁰⁴Pb vs. ²⁰⁶Pb/²⁰⁴Pb might be elucidated by a correlation between ²⁰⁶Pb/²⁰⁴Pb versus Pb/Ce, which might explain the isotopic variation in terms of loss or gain of lead in the source. I suspect something of that sort, because the Pb anomalies in the spidergrams shown by Nikogosian et al (2010) show an unusual extent of variability in their plot of "Group 3" lavas. Also, the isotope plots shown demonstrate that the Pb isotopes are the only ones that vary systematically, whereas the Nd and Sr isotopes are nearly uniform within analytical error. To me, that's another indication that addition/subtraction of lead, perhaps during subduction (?) was the main agent causing the isotopic variation. Moreover, the apparent Pb-Pb age of about 1.5 to 2. Ga obtained from the ²⁰⁷Pb/²⁰⁴Pb- ²⁰⁶Pb/²⁰⁴Pb array (after removal of two outliers afflicted by relatively large errors) seems to indicate that this lead mobility is ancient rather than recent. In any case, I suggest that the relationship between the new isotope data and the previously published chemical data has not been optimally exploited so far.

In summary, I recommend that this paper be revised significantly. These revisions should include a more substantial discussion of the relationship between isotopes and chemical composition, and I think the interpretation should be broadened to include a more serious discussion of alternatives to the model of recycling lower continental crust. I would leave any revisions regarding the focus on a metasomatized lithosphere up to the authors: I am not going to force them to abandon what seems to be their favorite model. After all, I cannot demonstrate that the metasomatic model is actually incorrect.

Finally, I would like to reiterate that the demonstration that rather precise isotope ratios for Pb, Sr, and Nd can be measured in melt inclusions constitutes a breakthrough, and this alone justifies publication in a high-visibility journal such as Nature Communications. Congratulations on this accomplishment!

Al Hofmann

Reviewer #2 (Remarks to the Author):

The article "Ancient recycled lower crust in the mantle source of recent Italian magmatism" deals the possible metasomatism of mantle wedge in the subduction zones revealed by isotopic data on melt inclusions in primary minerals of volcanic rocks. The approach is innovative because the new techniques permit to analyze very few amounts of trace elements. In addition, the whole rock isotopic compositions of volcanic rocks can be mask by contamination processes in shallow magmatic camera whereas the melt inclusions are unaffected by successive contamination and record the features of the mantle source. The paper is quite well written and the data support the conclusions. A point of weakness is section "results" which is too brief. More details on trace elements could be inserted.

I think this manuscript should be published with minor revisions evidenced in the remarks manuscript.

My main comments

- 1) Analytical data (in table) must also include trace elements as U, Th, Sr, etc.
- 2) An evaluation of CaO contents in melt inclusions would be desirable.
- 3) Isotopic compositions of melt inclusions into the two volcanoes must be presented separately.
- 4) The results are presented in a synthetic way; more details would be required.
- 5) Revised Figs. 3-4-5

Some suggestions are shown in the marked manuscript.

Annamaria Fornelli

Reviewers' comments:

Reviewer #1 (Remarks to the Author):

This paper presents new isotope data for melt inclusions in olivine crystals from one of the Italian magmatic provinces, which have been of great interest to geochemists for a long time, in part because of their extreme isotopic and chemical compositions.

The paper focuses on two young volcanic regions, namely Roccamonfina and Ernici volcanic provinces, which are part of the Roman Magmatic Province.

The paper introduces a major analytical advance to the study of melt inclusions, namely a full set of isotope data for the elements Pb, Sr, and Nd. Previous authors were unable to measure full $^{206}\text{Pb}/^{204}\text{Pb}$, $^{207}\text{Pb}/^{204}\text{Pb}$ and $^{208}\text{Pb}/^{204}\text{Pb}$ ratios because their instrumentation lacked the sensitivity to measure the abundance of the ^{204}Pb peak with sufficient precision. I think this advance alone justifies a publication in Nature Communications. Also, the paper is well written and should be of interest to much of the isotope geochemistry community.

Reply: We thank Prof Al Hofmann for his constructive and thorough review and very much appreciate that he acknowledges the importance of the new combined Sr-Nd-Pb MI data set to the isotope geochemistry community, and for his kind compliments towards the quality of the manuscript.

The interpretation of the data tends to follow previously established lines of reasoning, and it confirms and strengthens the evidence for an “exotic” source component tentatively identified in a previous paper by Nikogosian et al (2010). I generally do not try to argue about interpretations in a review of a paper, because I feel the authors are entitled to their own opinion, and my interpretive views may be wrong even though I am convinced I have the right answer. I will, in this case, make a partial exception to this general rule for myself, and I would like to encourage the authors to consider my arguments in a revised manuscript.

The authors make a case for the presence of ancient, recycled lower continental crust (LCC) in the source of the magmas, a component that is clearly identified only in the melt inclusions but is “buried” and not detectable in the whole rock compositions. I would agree that this is one possible model, which has in the past also been advocated for several so-called EM1-type oceanic basalts, with which the present rocks share several important characteristics. There is, however, an alternative explanation for the origin of at least some of these EM1-type basalts, which has also been explored extensively in the literature. This alternative is that EM1 isotopic characteristics (low $^{206}\text{Pb}/^{204}\text{Pb}$, high $^{208}\text{Pb}/^{206}\text{Pb}$, low $^{143}\text{Nd}/^{144}\text{Nd}$, intermediate $^{87}\text{Sr}/^{86}\text{Sr}$ ratios) can be explained by simple subduction and recycling of ancient pelagic sediments (e.g. Eisele et al. 2002, EPSL). This was always hard to distinguish from the alternative model invoking recycled lower continental crust, simply because the radiogenic isotopes allow both interpretations. For Pitcairn,

however, the issue has been largely resolved by a recent paper by Delavault et al. (PNAS 113, 12952, 2016) who showed evidence for sulfur isotopes affected by mass-independent fractionation, a phenomenon that is restricted to surface rocks older than about 2.4 Ga). Thus, it is pretty clear now that Pitcairn's EM1 signature is caused by recycled sediments, but this conclusion need not necessarily apply to all EM1 type basalts.

Reply: We appreciate that the reviewer suggests there are shared characteristics between some EM1-type oceanic basalts and the Italian Post-collisional lavas. We are aware that at rare OIB plume settings such as Pitcairn an isotopically similar 'EM1 component' has been identified in erupted lavas, and that derivation from ancient sediment recycling has been inferred. However, we sincerely wonder if making a comparison in the paper makes sense in light of the extremely different tectonic settings and thus melting processes at play. The Pitcairn ocean island is located far away from any continent or recent subduction zone whereas Italy is located on and surrounded by continental plates with multiple subducting plates resolved in the region. In contrast to the plume-related Pitcairn lavas that form from melting of an actively upwelling mantle potentially sampling the slab 'grave-yard' at great depths, the Italian magmas are produced within the shallow lithospheric mantle as a result of relatively recent subduction-related melting and there is no evidence of a plume-related influence. The presence of a plume containing ancient (>2 Ga), subduction-processed, and recycled sediments is thus not considered very likely beneath Italy which is why we did not make the link with EM1 in OIB settings in our previous version. We have, however, now added a section to discuss this possibility in lines 220-229 (note that all references to line numbers in these replies refer to the marked document).

Nevertheless, with the "Pitcairn experience" as background, I ask myself, which property of the present set of data requires them to have a lower crustal source rather than an ancient sedimentary source. It seems clear from the high CaO/Al₂O₃ ratios (Nikogosian, 2010) that this source component must be carbonate rich – not really a characteristic of the lower continental crust, but a perfectly reasonable component of a subducted sediment.

Reply: We agree that the carbonate rich characteristic is not typical for a lower crustal unit and therefore infer later secondary addition of this carbonate rich component either by metasomatism or by admixture of carbonate sediments. A short comment is made about this subject in line 216-219.

The component is particularly unradiogenic in Pb, a property that could characterize either a LCC component or an ancient sedimentary source.

Reply: The reviewer is correct, but is most likely only if the ancient sediment had been modified to loose U, freezing the Pb isotope ratio, which over time results in an unradiogenic signature. The loss of U can result from the sedimentation process itself but as can be seen from modern sediments with highly variable U/Pb this process is not consistent. Alternatively, the fractionation between U and Pb is suggested to happen in a subduction zone by dehydration from the slab as discussed in the early Pitcairn papers e.g. Woodhead and McCulloch 1989. This scenario requires the sediment to be subducted and 'processed' in an ancient >2 Ga subduction zone, to have resided at depth within the mantle for that time, to be subsequently entrained within the upwelling mantle and transported back up to the upper mantle to be involved in the melting. We

do not think this is a viable scenario considering the Italian subduction related tectonic setting and lack of evidence of the involvement of OIB. In Italy the involvement of old LCC or recently subducted old LCC sediments is more viable (see discussion in lines 220-229).

It is low in potassium, which might well be an indication of a sediment-derived primary carbonatite liquid.

Reply: We agree that the low potassium (and low sodium but high CaO) suggests derivation from a carbonatitic liquid, especially in combination with the observed enrichments in trace elements (Sr, U). The characteristics of this melt are very different from the known Italian subduction recycled Ca-rich sediments that have high K₂O and Na₂O. The coupled isotope and major and trace element characteristics suggest that the liquid is carbonatitic or derived from an apatite rich source, but the source must have lost U in an old event to have developed such unradiogenic Pb.

It has low Nb/U and Nb/U ratios, a property that might fit either source model. It is NOT particularly depleted in U or Th relative to other incompatible trace elements.

Reply: Indeed we agree that this is an important observation. We find that the component is rather U-enriched as can be seen from the negative relation between U concentration and Pb isotope compositions (Fig 4). We argue that the component must have experienced ancient U-depletion to result in the unradiogenic Pb, but that a recent mixing or metasomatic process has resulted in significant U (+Sr, REE) re-enrichment. We have now added a discussion to address this point in lines 216-219

It appears to have very slight negative Eu anomalies (Nikogosian, 2010), a property more in line with sediments than with lower crustal rocks.

Reply: The inclusions indeed have slight negative Eu anomalies, but we do not find a relation between the Pb isotope compositions and the Eu/Eu*. Note that the HKS host rocks also have Eu anomalies (see Figure below). The anomalies in host HKS lavas are, rather than plagioclase melt equilibration, explained as a characteristic of the melt source as there is no evidence for plagioclase on the liquidus of the melts (Hawkesworth and Volmer, 1979; Nikogosian et al., 2010). It is possible that the Eu anomalies in the MI's are inherited and controlled by mixing with melts from the bulk sediment source. Alternatively, the unradiogenic source may also have a negative Eu anomaly. Even though this would not be expected for a typical lower crust (Rudnick and Gao, 2003), it could have resulted from secondary metasomatic processes that led to the Ca- and TE enrichments.

Figure showing that there is no relation between the observed negative Eu anomalies and the Pb isotope characteristics of the melt inclusions. Host HKS rocks show similar Eu anomalies as the MI's.

It does have positive Sr anomalies, which would point toward a gabbroic and therefore lower crustal source, but which might also be generated by the carbonatitic nature of the primary liquids.

Reply: We agree that the enrichment in Sr is likely to result from the carbonatitic nature of the exotic melt component. We point out the enrichment in Sr and other TE's in lines 117-118, 160-162 and 215-218.

In summary, I find the evidence for a lower crustal source unconvincing, and I suggest that the sedimentary alternative is at least worth some discussion.

Reply: We agree that we cannot infer direct derivation from a LCC component based on the major and trace element characteristics. For this reason we argued that the LCC must have experienced carbonate metasomatism after the ancient orogenic event. Based on the suggestion by Al Hofman we now expanded on the scenario of subduction recycling of old LCC sediments from the hinterland and in this light consider admixing of carbonates present within the Adriatic sea to create the combined carbonatite-like TE enrichments and unradiogenic Pb compositions. This is now explicitly stated in lines 218 and 301-312.

The authors allow for recycled sediments in one part of the discussion, but they immediately specify that these sediments must be derived from a lower crustal source. I would insist that the source of such sediments must be old, but not necessarily derived from the lower crust. It seems to me that it would be hard to derive sediments directly from the lower crust. If the authors mean to derive these sediments from uplifted and eroded, originally lower crust, then this material has become part of the upper crust prior to erosion.

Reply: We agree and this is what we meant to imply. We now address this point more clearly in lines 301-312

Depending on the age of such crust, the Pb isotopes of sediments derived from it can be rather unradiogenic. For examples, the paper by Hemming et al. (2001,EPSL 184, 489-503) gives Pb isotopes for modern deep sea turbidites. Some of these turbidites have $^{206}\text{Pb}/^{204}\text{Pb}$ ratios as low as 16.8, although the vast majority of them do have ratios of 18 or greater. My point is that, if the sediments sample old rocks, they are likely to be rather unradiogenic in Pb. They do NOT have to come from the lower crust; they just need to be old.

Reply: The data-set in the Hemming and McLennan 2001 EPSL paper indeed shows that some rare sediments carry an unradiogenic signature (the vast majority of sediments has $^{206}\text{Pb}/^{204}\text{Pb} > 18.5$). Only sediments with $\epsilon\text{Nd} < 20$, Archaean in age (TDM > 2.3 Ga), have unradiogenic Pb compositions, because of inferred low U/Pb and Th/Pb in the crust at that time compared to the modern crust. Note that none of our melt inclusions have extreme Nd to support the involvement of this type of very old sediments with unradiogenic Pb (average in MI's $\epsilon\text{Nd} = -10 \pm 2$). In addition there is no Archaean basement present in the vicinity of the volcanoes studied here (the nearest craton is > 2000 km), which makes the involvement of such old craton-derived sediments unlikely. Because of the abundant presence of Variscan basement in the area (Fig. 1) that has unradiogenic Pb by metamorphic processes that lead to U loss we strongly prefer a scenario that involves recycling of ancient LCC. We thus now include a more comprehensive discussion on the LCC sediment recycling option as a potential mechanism to transport the unradiogenic Pb component into Italy's melt region (302-313 scenario 2, Fig. 5b).

There is another aspect of the manuscript that I find myself disagreeing with, namely the insistence that the recycled component, whatever it may be, has "metasomatized" the lithosphere, and it is ultimately the metasomatized lithosphere that melts. This view is extremely widespread among geochemists, but I find it is supported by remarkably little actual evidence. I don't think we know how to distinguish melts that ascend from the subducted slab directly to the surface from those that might first metasomatize the overlying lithosphere, which is then reactivated by some later time. And certainly, the tectonic situation in the Mediterranean region is extremely complicated, so I would find it hard to make a convincing case favoring one scenario over the other. Frankly, I think the model of "metasomatized lithosphere" has simply hardened into truth by frequent repetition in the literature rather than actual evidence. I am not going to insist on any reinterpretation of this aspect, but I do wish that the notion of the metasomatized lithosphere were not simply accepted as established truth.

Reply: We agree with the reviewer that we cannot physically prove the presence of a sediment metasomatized mantle wedge, however there are several lines of reasoning based on geochemical observations that support initial infiltration of fluids and melts from the slab and subsequent melting of the metasomatic sources.

First, Italian volcanics show an extremely wide range of major, trace element, and isotopic characteristics that are primarily of mantle origin. Quaternary rocks especially are characterized by large volumes of extremely enriched potassic and ultrapotassic rocks (Peccerillo et al., 20017). The observed extreme magmatic diversity requires the presence of a mineralogical heterogeneous

upper mantle (the presence of phlogopite and amphibole etc in various proportions, e.g. Conticelli et al., 2016). Geochronological, petrological and isotopic data convincingly suggest that these sources formed in multiple metasomatic events and that mantle metasomatic anomalies and heterogeneities increased with time.

Second, fluid and sediment transfer has been active since the onset of the Cenozoic subduction and this process inevitably led to impregnation of the mantle above the slab. By now, active subduction has, however, ceased under the Italian Peninsula and thereby also the active transfer of sediments and fluids from the Adriatic plate. The active volcanism over the last 1Ma thus likely does not directly result from fluids coming off the slab to induce the melting but rather from the inflow of hot material by slab tears and slab break-off (e.g. Spakman and Wortel, 2004).

Lastly, the fact that our exotic component inferred to be from LCC origin has a carbonatitic nature, rather different from the bulk potassic sediment infiltrated mantle, suggests that the unradiogenic Pb LCC component has been more recently infiltrated with carbonatitic melts or admixed with carbonate sediments to form such an enriched signature. It can, however, not be confirmed when this process took place.

A final point of this review concerns the treatment of the chemical composition of these basalts and their relationship to the newly measured isotopic compositions. I am writing this review from a place where I do not have access to the electronic supplement of Nikogosian's (2010) paper dealing with the chemistry of these melt inclusion, so I cannot make my own comparisons at the moment. I would consider it to be very helpful to the reader if the chemical data for those melt inclusions for which the authors have both isotopic and chemical data were listed again in an appropriate supplement, so that the reader who may want to look for alternative relationships does not have to dig through the literature and assemble a new Excel sheet to make plots.

Reply: We have created an Excel file that will serve as background data-set and contains all data that are discussed in the manuscript.

Specifically, I don't find the plots shown on Fig. 4 to be particularly instructive, because the absolute values of trace element concentrations can be affected by differences in the degree of melting, and I would hope to find better and more instructive correlations by plotting parameters such as $\text{CaO}/\text{Al}_2\text{O}_3$, Th/U , K/U , Nb/Th , Ce/Pb , Sr/Nd , Eu/Eu^* versus $^{206}\text{Pb}/^{204}\text{Pb}$ ratios in order to specify the composition and nature of the exotic crustal component. For example, the linear variation of the isotopic compositions, which is best displayed by a diagram of $^{208}\text{Pb}/^{204}\text{Pb}$ vs. $^{206}\text{Pb}/^{204}\text{Pb}$ might be elucidated by a correlation between $^{206}\text{Pb}/^{204}\text{Pb}$ versus Pb/Ce , which might explain the isotopic variation in terms of loss or gain of lead in the source. I suspect something of that sort, because the Pb anomalies in the spidergrams shown by Nikogosian et al (2010) show an unusual extent of variability in their plot of "Group 3" lavas.

Reply: We appreciate the comment and follow the reviewers suggestion to add diagrams of major and trace element ratio's versus the Pb isotope composition to better specify the composition and nature of the exotic component. We note that most of the ratios including the elements that we used in Fig 4 work equally well as the elemental concentrations. In a new version of Figure 4 we now include $\text{CaO}/\text{Al}_2\text{O}_3$, Sr/Nd , Th/U , U/Ce , La/Yb in addition to the Sr and La vs $^{206}\text{Pb}/^{204}\text{Pb}$ that

we keep (see new Fig 4). In addition we now also add diagrams of Sr/Nd and La versus $^{143}\text{Nd}/^{144}\text{Nd}$ as there are similar inverse correlations with trace-element concentrations and Nd isotopes to that expressed by the Pb isotopes. This expanded set of diagrams strongly supports the relation between the unradiogenic Pb composition (and more unradiogenic Nd) and the enriched carbonatitic nature of the exotic melt. The component is CaO-rich, K_2O - and Na_2O -poor and is enriched in Sr, La, U and other trace elements compared to the primitive component that has Pb isotope compositions similar to the bulk lavas. These relations are the basis for our argument that the exotic component is a LCC component affected by secondary carbonatitic metasomatism or was mixed with carbonate sediments. Note that we do not see relations with Pb concentration ratio's such as Pb/Ce (see below) that indicate addition or subtraction of Pb, nor do we see a relation with Eu that indicates a relation with plagioclase fractionation as was discussed above.

Figure showing the absence of a relation between Pb/Ce and the Pb isotope characteristic of the Roccamonfina and Ernici melt inclusions for which we have both isotope and trace element data.

Also, the isotope plots shown demonstrate that the Pb isotopes are the only ones that vary systematically, whereas the Nd and Sr isotopes are nearly uniform within analytical error. To me, that's another indication that addition/subtraction of lead, perhaps during subduction (?) was the main agent causing the isotopic variation.

Reply: Instead of addition or subtraction of Pb we suggest that ancient loss of the parent isotopes of Uranium and Th (but Th is less mobile) has played a major role in controlling the Pb isotope systematics. The fact that there is no relation with Sr and Nd isotopes means that the compositions of the exotic component (be it metasomatised LCC or sediment admixed LCC) and the bulk sediment+mantle component in the wedge cannot differ very much.

Moreover, the apparent Pb-Pb age of about 1.5 to 2. Ga obtained from the $^{207}\text{Pb}/^{204}\text{Pb}$ - $^{206}\text{Pb}/^{204}\text{Pb}$ array (after removal of two outliers afflicted by relatively large errors) seems to indicate that this lead mobility is ancient rather than recent.

Reply: We agree, as stated above, that U-removal must have been ancient. We suggest that U-removal took place during the Variscan orogeny or during even older events that occurred on the

northern margin of Gondwana. We use the similarity of isotopic compositions of LCC rocks from the nearby Massif Central to substantiate this inference.

In any case, I suggest that the relationship between the new isotope data and the previously published chemical data has not been optimally exploited so far.

Reply: We thank the reviewer for his thorough evaluation of potential relations between major and trace elements that argue for or against the two potential scenario's (LCC or old recycled sediments). We agree that we can be more comprehensive in our evaluation (especially because we now have less strict length constraints). We therefore included 4 more panels in Fig 4 and we have expanded the discussion to point out that the relations between Sr/Nd, La/Yb and U/Ce, Sr and REE vs Pb isotopes suggest potential coupled effects of degree of melting and source effects, i.e. mixing of melts from sources with different mineralogy, melting temperatures and melt productivity (lines 157-176)

In summary, I recommend that this paper be revised significantly. These revisions should include a more substantial discussion of the relationship between isotopes and chemical composition, and I think the interpretation should be broadened to include a more serious discussion of alternatives to the model of recycling lower continental crust. I would leave any revisions regarding the focus on a metasomatized lithosphere up to the authors: I am not going to force them to abandon what seems to be their favorite model. After all, I cannot demonstrate that the metasomatic model is actually incorrect.

Reply: We have made abundant revisions following the reviewers suggestions where we agreed with his views. We have also addressed the points where we disagree with his suggestion (e.g. involvement of an OIB component or old Archaean sediments) and we believe we have significantly improved the clarity and robustness of the argumentations in the paper.

Finally, I would like to reiterate that think the demonstration that rather precise isotope ratios for Pb, Sr, and Nd can be measured in melt inclusions constitutes a breakthrough, and this alone justifies publication in a high-visibility journal such as Nature Communications. Congratulations on this accomplishment!

Reply: We thank Al Hofmann for his enthusiasm and congratulations!

Al Hofmann

Reviewer #2 (Remarks to the Author):

The article "Ancient recycled lower crust in the mantle source of recent Italian magmatism" deals the possible metasomatism of mantle wedge in the subduction zones revealed by isotopic data on melt inclusions in primary minerals of volcanic rocks. The approach is innovative because the new techniques permit to analyze very few amounts of trace elements. In addition, the whole rock

isotopic compositions of volcanic rocks can be mask by contamination processes in shallow magmatic camera whereas the melt inclusions are unaffected by successive contamination and record the features of the mantle source. The paper is quite well written and the data support the conclusions. A point of weakness is section "results" which is too brief. More details on trace elements could be inserted.

I think this manuscript should be published with minor revisions evidenced in the remarks manuscript.

Reply: We thank Annamaria Fornelli for her positive and helpful review. We have replied to all her comments in the marked manuscript that also tracks our changes according to the comments and suggestions of both reviewers.

My main comments

1) Analytical data (in table) must also include trace elements as U, Th, Sr, etc.

Reply: Following a similar request from Al Hofmann we now include the full data-set as a Background data-set Excel file.

2) An evaluation of CaO contents in melt inclusions would be desirable.

Reply: The CaO contents can now be found in the background data set. The relations between CaO and other major and trace element ratios have been shown and discussed extensively in Nikogosian and Van Bergen 2010. We choose here to display ratios of CaO/Na₂O (Fig 2) and CaO/Al₂O₃ (new figure 4) to evaluate the relative enrichment and depletions in these elements. We believe that we now present sufficient information for the reader to see that we have fully evaluated the relationships.

3) Isotopic compositions of melt inclusions into the two volcanoes must be presented separately.

Reply: We have changed the text and figures to now discuss and display the Roccamonfina-Ernici data separately.

4) The results are presented in a synthetic way; more details would be required.

Reply: We have deliberately kept the results section short to avoid repetition of the Nikogosian and Van Bergen article, which describes the petrography and major and trace element data in detail. We have, however, expanded the results section to now discuss the two volcanoes separately and to include more details with respect to the relations between major and trace elements and Pb isotope characteristics.

5) Revised Figs. 3-4-5

Reply. I presume the reviewer meant to say revise Fig 2, 3 and 4 to display the data for the two volcanoes separately. We have followed her suggestion and have revised the figures accordingly.

Some suggestions are shown in the marked manuscript.

Reply: We have addressed all comments and suggestions in the track changes document.

Annamaria Fornelli

Reviewers' comments:

Reviewer #3 (Remarks to the Author):

In most respects this is a very good manuscript. It contains very valuable data obtained through analysis of melt inclusions using innovative analytical techniques. These data potentially provide important constraints on the origin of unusual volcanic rocks in central Italy.

I do, however, have some difficulties in accepting the interpretations. I recognize that some parts of the sub-continental lithospheric mantle are metasomatised and that in some cases a component from this source might be introduced into magmas that subsequently rise to the surface, but I believe that this mechanism is invoked far too often in the literature, commonly with little to no real justification.

In the case of the Quaternary volcanic rocks from Italy, a good case can be made that their compositions reflect their derivation in a subduction setting and that they contain material from lithosphere (the mantle wedge) impregnated by fluids from subducting slabs. Influx of fluid or melt from subducting pelagic and carbonate sediment can explain much of the geochemical characteristics of these rocks, as invoked by many authors who have published on this subject. The critical question, though, is how to explain the origin of what the authors of the present manuscript call the "exotic" component. The authors propose that this component represents lower crustal material that was transported into the subcontinental lithosphere and then, somehow, was incorporated into the magmas. It is this part of the manuscript that I believe requires some work. Although the component does indeed share some characteristics with rocks from the lower continental crust (low and variable Pb isotope ratios combined with low $^{143}/^{144}\text{Nd}$), it also has some unusual aspects. Of these, the most striking is the relatively high, but remarkably constant, $^{87}/^{86}\text{Sr}$ ratio. In Fig. 3D we see a significant variation in $^{143}/^{144}\text{Nd}$ at near-constant $^{87}/^{86}\text{Sr}$ in the melt inclusions. This chemical trend contrasts markedly with the large variation of $^{87}/^{86}\text{Sr}$ in the lower crustal granulites that are also shown in the figure. In Fig 3B, we see a large range in $^{206}/^{204}\text{Pb}$ and near-constant $^{87}/^{86}\text{Sr}$ which also contrasts markedly with covariation of the isotope ratios in the granulites. This is to be expected – there is no reason why material from the lower crust should have uniform $^{87}/^{86}\text{Sr}$.

To explain the near-constant $^{87}/^{86}\text{Sr}$, the authors propose that the "exotic" component had the same ratio as the dominant component in the mantle source of the lavas. In other words, the sediment-derived subduction component in the mantle source had identical $^{87}/^{86}\text{Sr}$ to that of older continental crust. Well, maybe, but given the big range in rocks and mantle compositions displayed in Fig 3D, such a coincidence seems very unlikely. Normally one would expect some correlation between Pb, Nd and Sr isotopic compositions controlled in part by magmatic/sedimentary processes and in part by variable U loss \pm Sr loss (???) during granulite-facies metamorphism.

Perhaps some other process has pinned the $^{87}/^{86}\text{Sr}$ ratios in the lavas. On page 6, the authors state that strong CO_2 emissions from Italian volcanoes are due to a high proportion of carbonate in subducted lithologies. Although this may be partly correct (e.g. Avanvinelli et al 2018, *Geology* 46, 259), there is strong evidence that the strong CO_2 emissions result mainly from the assimilation of crustal limestones that directly underlie the volcanoes (e.g. Iacono-Marziano et al. 2009, *Geology* 37, 319). I wonder whether assimilation of limestone could not have controlled the Sr isotopic compositions of the lavas. In Fig. 3D, the authors plot the compositions of "Umbria carbonates" (but do not give the source for these compositions) whose $^{87}/^{86}\text{Sr}$ is higher (.711-.712) than measured in the melt inclusions (.710) and that of most limestones in the region (.707). Is there any evidence of carbonate sedimentary rocks with $^{87}/^{86}\text{Sr} = 0.710$ in the region? I'm fully aware of that the low Sr content of limestone limits the influence of assimilation of this rock type, but, as Iacono-Marziano et al point out, very large amounts of carbonate can be assimilated having a major influence on the composition of the contaminated magma. Addition of

carbonate adds only Ca and Mg, which leads to crystallization of more Ca-cpx that bring the composition back to nearly where it was, and to the release of abundant CO₂ (that monitored in the compositions of volcanic gases). Could fluxing of CO₂-rich fluid through the magma have buffered the Sr isotopic composition? Whatever the process, the authors must find a reasonable explanation for the peculiarly constant 87/86Sr if they wish to advocate a lower-crust component in the source.

Another issue is the mechanism that introduced lower crustal material first into the lithospheric mantle and then into the magmas. The (overlong?) section entitled "Provenance of the LLC component ... " describes various processes that might have transported segments of lower crust into the mantle and we read phrases like "melts extracted from the transported segments metasomatised the lithosphere". Exactly how did this happen? And more to the point, how was this component, and the other component that was the main source of the lavas, extracted from the lithosphere to produce the binary trends illustrated in Fig 4? Why just these two components and nothing else? Furthermore, any process that extracts a small amount of material from a larger reservoir – partial melting or some sort of reaction – will progressively change trace element ratios in the extracted material. These effects would disturb the trends of Fig 4. On line 261 it is proposed that erosion of exhumed LCC lithologies produced sediments that were subsequently subducted then added to the lithospheric mantle. I find it difficult to imagine that such a process could introduce the "exotic" component into the source without adding material with more normal upper crustal compositions.

This might all sound very picky, but these are the sorts of questions that should be asked before advocating a "metasomatised mantle source". Too often the subcontinental lithospheric source is just considered a convenient storage site into which material can be added or extracted, with no consideration about how this is done.

My preferred interpretation would be that magmas from the mantle wedge stalled in the lower crust. There they assimilated some wall rock and some of this material was trapped in early crystallizing olivine. Then, higher in the crust, limestone assimilation buffered the Sr isotopic compositions.

Whether the authors are willing to accept, or even consider, this interpretation is of course up to them, and to the editor.

Nicholas Arndt
Rennes, 25/12/2018

Response to reviewers

Reviewer #3 (Remarks to the Author):

In most respects this is a very good manuscript. It contains very valuable data obtained through analysis of melt inclusions using innovative analytical techniques.

Reply: We thank Prof. Arndt for acknowledging the quality of the manuscript and the novelty of the Sr-Nd-Pb isotope data on melt inclusions that we present.

These data potentially provide important constraints on the origin of unusual volcanic rocks in central Italy. I do, however, have some difficulties in accepting the interpretations. I recognize that some parts of the sub-continental lithospheric mantle are metasomatised and that in some cases a component from this source might be introduced into magmas that subsequently rise to the surface, but I believe that this mechanism is invoked far too often in the literature, commonly with little to no real justification.

In the case of the Quaternary volcanic rocks from Italy, a good case can be made that their compositions reflect their derivation in a subduction setting and that they contain material from lithosphere (the mantle wedge) impregnated by fluids from subducting slabs. Influx of fluid or melt from subducting pelagic and carbonate sediment can explain much of the geochemical characteristics of these rocks, as invoked by many authors who have published on this subject. The critical question, though, is how to explain the origin of what the authors of the present manuscript call the "exotic" component. The authors propose that this component represents lower crustal material that was transported into the subcontinental lithosphere and then, somehow, was incorporated into the magmas. It is this part of the manuscript that I believe requires some work. Although the component does indeed share some characteristics with rocks from the lower continental crust (low and variable Pb isotope ratios combined with low $^{143}/^{144}\text{Nd}$), it also has some unusual aspects. Of these, the most striking is the relatively high, but remarkably constant, $^{87}/^{86}\text{Sr}$ ratio. In Fig. 3D we see a significant variation in $^{143}/^{144}\text{Nd}$ at near-constant $^{87}/^{86}\text{Sr}$ in the melt inclusions. This chemical trend contrasts markedly with the large variation of $^{87}/^{86}\text{Sr}$ in the lower crustal granulites that are also shown in the figure. In Fig 3B, we see a large range in $^{206}/^{204}\text{Pb}$ and near-constant $^{87}/^{86}\text{Sr}$ which also contrasts markedly with covariation of the isotope ratios in the granulites. This is to be expected – there is no reason why material from the lower crust should have uniform $^{87}/^{86}\text{Sr}$.

Reply: The reviewer questions our interpretation that the “exotic” component represents lower continental crust (LCC) based on the ‘constancy’ of Sr isotopes at variable Pb isotope ratios. Referring to general heterogeneity in lower crustal granulites, he expects the variability in Sr isotope composition to be of a similar extent as the Pb isotopes, and the MIs to show similar variability as the plotted Massif Central data in Fig. 3.

In contrast to his perception, we need to emphasize here that we do not attribute the Pb isotope variability in the melt inclusions (MIs) to 'low and variable Pb' in the LCC component. Instead, the variability that we see in the MIs reflects mixing of variable contributions from isotopically distinct components during melt mixing and entrapment. The components themselves are interpreted to have relatively invariable Pb and Sr isotope compositions. We thus do not invoke a range of granulites with variable isotopic signatures to explain the Roccamonfina-Ernici MI results. Instead, we infer the presence of a single LCC endmember component with unradiogenic Pb in the mantle source, along with another, more abundant sedimentary/UCC-derived component. Our comparison with the Massif Central granulites in Fig. 3 is merely to demonstrate that lower crustal rocks with characteristics close to this 'exotic' component exist within Europe, not to see if the isotopic diversity in LCC granulites matches that of the MI. The Massif Central granulites reported in Fig 3 derive from 3 localities (~100 km apart) with highly variable paragenesis (meta igneous and meta sedimentary; Downes et al., 1990; Downes et al., 1991 and references therein), which is reflected in the large range of isotopic compositionc.

Note that we argue that owing to mixing processes melt inclusions trapped in olivine will not directly record the composition of the exotic source component. The Sr isotope data of mixed melts do not necessarily need to be similarly variable as the Pb isotopes, since the mixed signatures will depend on the difference in elemental concentrations and Sr isotope compositions of the melt sources (see further discussion below). The mixed nature of the MI, precludes pinpointing the endmember component composition exactly. Our data, however, define relationships that suggest that the component has more unradiogenic Pb and Nd isotope compositions than the Massif central granulite rocks.

To avoid possible misunderstanding that the isotopic variability in our MI would correspond to that of all Massif Central LCC rocks we changed the wording in the text in lines 255-260.

To explain the near-constant $87/86\text{Sr}$, the authors propose that the "exotic" component had the same ratio as the dominant component in the mantle source of the lavas. In other words, the sediment-derived subduction component in the mantle source had identical $87/86\text{Sr}$ to that of older continental crust. Well, maybe, but given the big range in rocks and mantle compositions displayed in Fig 3D, such a coincidence seems very unlikely. Normally one would expect some correlation between Pb, Nd and Sr isotopic compositions controlled in part by magmatic/sedimentary processes and in part by variable U loss \pm Sr loss (???) during granulite-facies metamorphism.

Reply: The reviewer suggest here once more that he expects the variability shown by the melt inclusions to reflect the variability of the source directly, and thus that he would expect the covariations to result from, for example, granulite forming processes. Even though it may seem coincidental, our reasoning to suggest the exotic source has a similar $^{87}\text{Sr}/^{86}\text{Sr}$ is based on the combined trace element and isotope data of MIs that suggest two-component binary mixing.

Mixing curves in two-isotope diagrams are defined by the isotope compositions of the endmembers, on the concentrations of the elements (e.g., Langmuir et al., 1978) and on the extent and fashion of mixing (e.g., Rudge et al., 2013). In a simple scenario where two melt components mix, we obtain straight to highly curved hyperbolic lines. If concentration ratios in the endmembers are very different the mixing trend will be strongly curved. That is unless we sample

‘the flat part’ of the curve. In the following we demonstrate the reasoning behind the statements in the manuscript that the reviewer refers to (lines 187-189).

Figure R1 below shows a simple mixing model for melt pairs with trace element concentrations similar to our two extreme melt inclusions (R-139 with unradiogenic $^{206}\text{Pb}/^{204}\text{Pb}$, Sr = 2820 ppm; Pb = 15 ppm; E-363 with $^{206}\text{Pb}/^{204}\text{Pb}$ similar to the host lava, Sr = 349 ppm; Pb = 15 ppm). We keep the isotopic composition of one end-member melt composition constant (composition of MI E363) and arbitrarily fix the $^{206}\text{Pb}/^{204}\text{Pb}$ of the second component at 15 while varying the Sr isotope composition. The relatively constant Sr isotope compositions of our MI (red circles) is only produced if the two endmember melts have a similar Sr isotope composition.

Figure R1. Simple binary mixing relations between melts of MI ERN-363 and a component with variable $^{87}\text{Sr}/^{86}\text{Sr}$ but Pb-Sr concentrations of R-139, compared to the MI compositions. Only mixing with an endmember that has similar $^{87}\text{Sr}/^{86}\text{Sr}$ to our inclusions generates the observed narrow range in Sr isotope compositions.

The exact Nd-Pb isotope composition of the exotic component that contributed to the melt inclusions is difficult to constrain because the isotopic mixing relationships depend on the isotopic composition and concentrations of the end-member components that in turn depend on the melting conditions, the degree of mixing and at what stage melts are trapped or extracted (Koornneef et al., 2012; Rudge et al., 2013). For this reason any firm conclusion about exact compositions or relative contributions of each component to the inferred mixing trend remains elusive. Our general inferences are, however, robust.

Using the binary mixing example presented above, we have extended the text in lines 180-189 to more clearly explain our claim that the exotic component must have similar $^{87}\text{Sr}/^{86}\text{Sr}$ as the sediment derived source.

Perhaps some other process has pinned the $^{87}/^{86}\text{Sr}$ ratios in the lavas. On page 6, the authors state that strong CO_2 emissions from Italian volcanoes are due to a high proportion of carbonate in subducted lithologies. Although this may be partly correct (e.g. Avanzinelli et al 2018, *Geology* 46, 259), there is strong evidence that the strong CO_2 emissions result mainly from the assimilation of crustal limestones that directly underlie the volcanoes (e.g. Iacono-Marziano et al. 2009, *Geology* 37,

319). I wonder whether assimilation of limestone could not have controlled the Sr isotopic compositions of the lavas.

Reply: We concur that part of the observed CO₂ outgassing from volcanoes and diffuse sources in Italy is locally derived from interactions of mantle derived magmas with crustal carbonates. However, we explicitly exclude limestone assimilation in the crust as explanation for the Sr isotopic composition of the MI for a number of reasons:

The rationale behind the methodology used in our study is to specifically select primitive olivine that crystallised and entrapped MI at deep levels. This ensures that any effects from crustal assimilation are avoided/minimised (e.g., see Saal et al., 1998). The analyzed inclusions and their host olivine were thoroughly characterised in terms of major and trace element compositions. This work is especially aimed at determining the nature of the mantle source and partial melting controls, as outlined in Nikogosian and van Bergen, 2010.

Discussion of the possible role of crustal assimilation versus mantle enrichment processes below Italy stems back to the 1970s (Hawkesworth and Vollmer, 1979; Rogers et al., 1985). Consensus was reached after two decades of work in the late 1990s (e.g., Conticelli, 1998): (i) At some locations (e.g. Alban Hills, Vesuvius, San Venanzo) whole rock lavas show clear evidence of assimilation within the crust. This assimilation has obvious effects on the primary chemical and isotopic characteristics of the lavas (e.g., Peccerillo, 2017), (ii) in contrast, Sr-Nd-Pb isotope compositions of high-MgO lavas erupted in the majority of volcanic centres in mainland Italy are not significantly affected by assimilation and are inherited from their mantle source (Conticelli et al., 2015; Peccerillo and Lustrino, 2005). The petrography and geochemistry of the host lavas and the constituent minerals from Roccamonfina – Ernici have been extensively described in the literature (Boari et al., 2009; Conticelli et al., 2009a and many refs therein) and do not point towards a major role of crustal assimilation at these locations.

Textural and compositional characteristics of the host lavas, constituent minerals and melt inclusions in our study are in agreement and testify against a crustal assimilation effect:

1) Absence of textural evidence for assimilation

Assimilation of limestone will result in reactions with the magma that affect the petrology and petrographic textures of the volcanic products. This is demonstrated for example in high-potassium magmatism of the Alban Hills by the presence of calcite inclusions within phenocrysts, by sparry calcite in the groundmass, as well as late veinlets along grain boundaries and fractures (Gaeta et al., 2006; Peccerillo et al., 2010). Furthermore, limestone assimilation will result in changing compositions of phenocrysts (olivine, cpx) crystallizing from the (contaminated) melt and/or unusual zoning (e.g. Ca-rich rims/overgrowths), or in corroded phenocrysts (e.g., K-feldspars with granular hypidiomorphic textures at Alban Hills). Finally, reactions with limestone would liberate excess CO₂ (Iacono-Marziano et al., 2009) potentially to be included in crystallizing phenocrysts. The Roccamonfina-Ernici host rocks investigated in this study (Rocc-7 and Ern-4) do not show any of such petrographic evidence for limestone assimilation; i) the rocks do not contain calcite in any form, nor any calc-silicate-rich reaction products (such as described for Vesuvius); ii) the olivine phenocrysts with trapped primary MIs do not show any unusual textural features; iii) primary olivine does not contain CO₂ rich fluid inclusions.

2) Primary compositions of host olivine

Assimilation of limestone would cause chemical modifications of lavas and constituent minerals. Olivine forsterite content ($\text{Mg}/(\text{Mg}+\text{Fe}) * 100 = \text{Fo mol}\%$) is a measure of how primitive or evolved a magma was when the olivine crystallised. The higher the Fo content, the more primitive the host magma. This is because fractional crystallization of olivine and clinopyroxene during magma evolutions will result in lowering the MgO while FeO increases. Limestone assimilation will result in a decrease in the Fo in olivine rims while CaO would increase (Gaeta et al., 2009; Lustrino et al., 2016). We observed these compositional effects at San Venanzo (See Fig. R2) but our Roccamonfina-Ernici olivines lack any such Fo zonation towards the rim (e.g., See Fig. R3). They have Mg#'s 88 – 90 mol.% typical for primary olivine. Hence, we agree with many previous workers (Boari et al., 2009; Conticelli et al., 2009 and several refs therein) that there is no evidence for significant crustal assimilation in the primitive lavas at Roccamonfina-Ernici.

Fig. R2. Compositional profiles of olivine phenocrysts affected by assimilation in San Venanzo (Ersoy et al., GCA, accepted).

Fig. R3. Fo-rich olivines from Roccamonfina-Ernici do not show abnormal compositional zonation.

3) Primary compositions of trapped melts

If limestone assimilation had occurred, it should be evident in the composition of the MIs. In addition to enrichment in CaO and relative depletions in SiO₂ (and other major components), assimilation would lower the concentrations of virtually all incompatible trace elements (ITE). The Roccamonfina/Ernici MI show the opposite. The inclusions carrying the unradiogenic Pb component is enriched in many ITE (Fig. 4 in manuscript). Compared to Italian Mesozoic limestone from the Apennines (Conticelli et al., 2009b), the ITE concentrations of Roccamonfina/Ernici MI are 10-100 times larger. Also, the ITE distribution pattern is very different from that of melts from the San Venanzo centre that experienced carbonate assimilation (Fig. R4).

Fig R4. Comparison of primary HKS melt inclusions from Roccamonfina/Ernici with Apennine Limestone. Trace element patterns of Roccamonfina-Ernici melt inclusions do not comply with limestone assimilation. Note that melt inclusions that assimilated limestone (San Venanzo, panel on the right) show much lower trace element contents. Data from Nikogosian et al., 2013.

Thus, from this textural and compositional evidence we can exclude the possibility that our MI compositions were influenced by limestone assimilation in the crust. Note that assimilation of crustal material was not considered to be relevant by reviewers #1 or #2 in the initial review round.

To address the issue more explicitly we adjusted and added text at the start of the section 'Melt supply from distinct mantle components' (lines 116-121). We have furthermore added the two references Avanzinelli et al., 2018 and Iacono-Marziano et al. 2009 mentioned by the reviewer to the discussion.

In Fig. 3D, the authors plot the compositions of "Umbria carbonates" (but do not give the source for these compositions) whose $^{87}\text{Sr}/^{86}\text{Sr}$ is higher (.711-.712) than measured in the melt inclusions (.710) and that of most limestones in the region (.707). Is there any evidence of carbonate sedimentary rocks with $^{87}\text{Sr}/^{86}\text{Sr} = 0.710$ in the region?

Reply: We thank the reviewer for spotting a typo in Fig. 3D and for drawing attention to the isotopic difference between our MI and limestone in the region. The field labelled "Umbria carbonates" should have been labelled "Umbria carbonatites". The data source is the same as that of the Italian lavas (Lustrino et al., 2011). These Umbria rocks in the internal Apennines are magmatic in nature. As to the question if there are any sedimentary carbonate rocks with $^{87}\text{Sr}/^{86}\text{Sr} = 0.710$ in the region, the answer is no. Limestone compositions reported in the literature define a relatively narrow range between $^{87}\text{Sr}/^{86}\text{Sr} = 0.707$ and 0.708 (Frijia et al., 2015; Melluso et al.,

2003). Assimilation with typical Italian limestone would thus not explain the Sr-isotopic compositions of the MI. In Fig R5 we plot the Sr-Nd isotope variability of the MI together with a binary mixing model between inclusion ERN-363 (least affected by the exotic component, also used in Fig. 4) and a typical Italian limestone with $^{87}\text{Sr}/^{86}\text{Sr} = 0.707453$, $^{144}\text{Nd}/^{143}\text{Nd} = 0.511824$; Sr = 645 ppm; Nd = 9 ppm (Conticelli et al., 2009a). This simple model underscores the mismatch between the MI compositions and a mixing trend with Italian crustal limestone (Fig R5). Unfortunately, no Pb isotope data are available for Italian limestone but it is not expected to have unradiogenic Pb isotope compositions, being generally marked by relatively high U/Pb ratios.

To avoid confusion in Fig 3, we took out the fields for carbonatites (Umbria 'carbonates' and Tanzania carbonatites) that we do not discuss in the text and are thus redundant.

Fig. R5. Binary mixing model between a typical Italian limestone and melt inclusion ERN-363. Tick marks represent 10% fractions.

I'm fully aware of that the low Sr content of limestone limits the influence of assimilation of this rock type, but, as Iacono-Marziano et al point out, very large amounts of carbonate can be assimilated having a major influence on the composition of the contaminated magma. Addition of carbonate adds only Ca and Mg, which leads to crystallization of more Ca-cpx that bring the composition back to nearly where it was, and to the release of abundant CO₂ (that monitored in the compositions of volcanic gases). Could fluxing of CO₂-rich fluid through the magma have buffered the Sr isotopic composition?

Reply: The reviewer correctly states that Italian Mesozoic limestones tend to have (much) lower Sr contents compared to mantle derived melts of the Roccamonfina-Ernici centres. In addition, the Sr isotopic ratios are also lower (as Fig R5 illustrates). This rules out any buffering of Sr isotopes by interaction of mantle derived melt with limestone in the crust. Also, at the relatively shallow depth where limestone assimilation would occur, CO₂-rich fluid would exsolve. However, phenocrysts in our samples do not contain CO₂-rich fluid inclusions to support fluxing by CO₂-rich fluids. Finally, as has been documented for Italian high-K magmas, the mineral chemistry of clinopyroxene and olivine would show compositional effects from limestone assimilation (e.g., Gaeta et al., 2006; Di Rocco et al., 2012) but, as mentioned above, such evidence is lacking in our samples.

We thus conclude that: 1) assimilation did not play a major role in modifying primitive magma compositions at Roccamonfina-Ernici; 2) our MI represent the most primary melts in the system; 3) the compositional variability in the MI must reflect heterogeneity in the mantle source.

Whatever the process, the authors must find a reasonable explanation for the peculiarly constant $^{87}\text{Sr}/^{86}\text{Sr}$ if they wish to advocate a lower-crust component in the source.

Reply: In Fig R6 we show, as an illustration, binary mixing curves between mantle-derived endmembers with known compositions. We use a) EM1, or b) Lewisian Gneiss as recycled LCC components and indicate mixing with the bulk host lava composition. Mixing between EM1 (Armienti and Gasperini, 2007) and the Rocc-7 Host lava fails to explain the co-variability in the inclusions. However, mixing between ancient lower crustal granulites from Scotland (Lewisian gneiss, Chapman and Moorbath, 1977; Dickin, 1981), does mimic the observed trend even if the Sr concentration and isotope composition is significantly lower compared to the most extreme inclusion compositions. Note that the mixing line approaches a flat curve, with relatively invariable Sr isotope compositions, when mixing approaches a >50% contribution from the host. Based on this oversimplified solid mixing model we argue that in a more realistic preferred scenario where we mix small amounts of partial melts from a LCC component in the Roccamonfina-Ernici mantle wedge (see our discussion above at p2-3 and Fig. R1) it will be more easy to account for the relatively constant $^{87}\text{Sr}/^{86}\text{Sr}$ at variable $^{206}\text{Pb}/^{204}\text{Pb}$ composition. This is because element concentrations in the partial melts have a large effect on how mixing relations are expressed in two-element isotope diagrams. As discussed in lines 153-174, the high TE concentrations in the unradiogenic Pb component can either be an inherent feature of the source component or result from small degree melting. A mixing process involving TE-enriched partial melts (rather than the solid) would drastically decrease the required contribution of the LCC component to a more realistic abundance (<5%).

We now discuss the constancy of $^{87}\text{Sr}/^{86}\text{Sr}$ controlled by melt mixing more extensively in lines 179-190. In lines 277-280 we discuss an additional scenario where the apparent enrichment and constant Sr isotope composition results from admixture of subducted Adriatic marine carbonates to a LCC sediment component in the context of provenance of the LCC component.

Fig. R6. The variability in Roccamonfina/Ernici melt inclusions compared to binary mixing models of solids (Lewisian gneiss LCC – host lava and EM1 – Host lava) and melts (MI R-139 and E-363 compositions and a hypothetical LCC melt, see also Fig R1). Although the mixing curve between Lewisian gneiss and the host lava explains the invariable Sr isotope composition of the MIs we infer mixing of partial melts, suggesting that the LCC should have similar $^{87}\text{Sr}/^{86}\text{Sr}$ compared to the host lava.

Another issue is the mechanism that introduced lower crustal material first into the lithospheric mantle and then into the magmas. The (overlong?) section entitled "Provenance of the LLC component ... " describes various processes that might have transported segments of lower crust into the mantle and we read phrases like "melts extracted from the transported segments metasomatised the lithosphere". Exactly how did this happen?

Reply: The Pb isotopes strongly suggest involvement of an ancient LCC component in the magma source (a new and unexpected discovery for Italian potassic magmatism) and we thus deem it crucial to explore how such material could have been recycled into the mantle below Italy within the regional geodynamic context. For this purpose, we use the available literature to assess where LCC material is present in the region and conclude that there are multiple routes along which it could have been transported into the mantle domain below the volcanic centres. The passage the reviewer refers to here (lines 266-269) describes a scenario referred to by Lustrino (2005) to explain the compositional variability observed in Sardinia lavas. Lustrino argued that delamination and subsequent sinking of lower crustal segments could have resulted in metasomatism of the overlying mantle by melts from the sinking segment. We use this suggestion and existing geodynamic reconstructions of the Western Mediterranean realm in support of potential recycling of LCC. However, as we state in the manuscript (lines 314-318) we do not favour any of the proposed options nor do we claim to know the exact physical mechanism and form in which LCC was entrained, travelled and ended up in the magma source. We, however, hope to with this new finding and the inferred potential scenarios, encourage the reader and future researchers to think

about the potential large-scale geodynamic processes that can explain the geochemical data. We intend to motivate geochemists and geodynamicists to find ways to confirm the geodynamic processes that control elemental and isotope fluxes into the mantle.

We think that the length of the section is appropriate and have not adjusted the text.

And more to the point, how was this component, and the other component that was the main source of the lavas, extracted from the lithosphere to produce the binary trends illustrated in Fig 4? Why just these two components and nothing else? Furthermore, any process that extracts a small amount of material from a larger reservoir – partial melting or some sort of reaction – will progressively change trace element ratios in the extracted material. These effects would disturb the trends of Fig 4.

Reply: Here, the reviewer raises an issue that indeed is intriguing. We fully agree that partial melting and mixing of partial melts during melt extraction will progressively change the trace element ratios in extracted melts (see e.g. the discussion and model in Koornneef et al., 2010 for the Iceland plume scenario or Nikogosian and van Bergen 2010 for Roccamonfina-Ernici). As discussed in the manuscript, trace element concentrations and inter-element variations in the extracted melts are a function of the source composition as well as the mineral assemblage, which controls the melting temperature, productivity and thus the degree of partial melting as a function of pressure and temperature. Melting of different source components can initiate at different depths depending on the fusibility of the mineralogy (e.g., Stracke et al., 2003).

The binary mixing relations that we see between the Pb isotope compositions and trace element contents and ratios suggest entrapment of mixed melts from 2 distinct components during increasing degrees of partial melting. It follows from the observed geochemical relations that the Ca-rich LCC component contributes more to the bulk primary melt at small degrees of partial melting, presumably deeper in the system. This LCC component then becomes progressively diluted by mixing with the bulk melt during continued melting of the mantle wedge/ lithospheric mantle (decreasing La/Yb, Sr/Nd, U/Ce but increasing $^{206}\text{Pb}/^{204}\text{Pb}$ and $^{143}\text{Nd}/^{144}\text{Nd}$; Fig. 4). Note again that we consider this 'bulk melt' to be derived from portions of the mantle that had been metasomatised by the subducted sedimentary component that is generally inferred to have produced the sources of Italian K-rich magmas. Early crystallization of olivine phenocrysts from melt volumes prior to complete homogenisation explains the entrapment of primitive melts with a range of compositions.

Following the reviewer's encouragement we expanded the section on the relationships between trace element and isotope systematics in lines 153-173.

On line 261 it is proposed that erosion of exhumed LCC lithologies produced sediments that were subsequently subducted then added to the lithospheric mantle. I find it difficult to imagine that such a process could introduce the "exotic" component into the source without adding material with more normal upper crustal compositions.

Reply: The reviewer seems to suggest here that we ignore a role for 'normal' upper crustal sediment recycling and would solely consider subduction-mediated addition of sediment from an exhumed LCC block. This suggestion is incorrect. In lines 128-134 we describe that melting of a

mantle modified by a mixed carbonate-pelitic sedimentary component derived from upper continental crust (UCC) best explains the geochemical characteristics of Roccamonfina-Ernici HKS bulk lavas. However, aside from the major role of these upper crustal sediments, we argue that sediments eroded from exhumed LCC, could potentially have introduced the minor exotic component into the mantle source in a similar subduction-related fashion. Note that this recycling of ancient sediments derived from the hinterland, was suggested as an option by Reviewer #1. We had therefore at the first round of revisions expanded the discussion to include this as a potential mechanism and thus choose to leave the text unchanged.

This might all sound very picky, but these are the sorts of questions that should be asked before advocating a "metasomatized mantle source". Too often the subcontinental lithospheric source is just considered a convenient storage site into which material can be added or extracted, with no consideration about how this is done.

Reply: We fully agree with the reviewer that simply ascribing a geochemical signature to an exotic source deep in the lithosphere is unsatisfactory and thus specifically do not do this. Our study provides evidence for the involvement of an old lower crustal component from the isotopic compositions of mafic MI trapped in early crystallized olivine. Based on our specific methodology and observations, we argue that the LCC component resides in the magma source. Combined with extensive support from the literature (Avanzinelli et al., 2009; Conticelli et al., 2015; Peccerillo, 1999; Peccerillo, 2017) we consider mantle metasomatism as a viable mechanism for creating mantle heterogeneity by element transport in the Italian subduction setting. At the same time we point out that the Western Mediterranean setting offered many opportunities for introduction of an LCC component in the mantle wedge/lithosphere below mainland Italy, but concede that we currently have insufficient constraints to distinguish between the various geodynamic scenarios potentially available. Further work is certainly required to trace the exact provenance and recycling pathways in detail.

My preferred interpretation would be that magmas from the mantle wedge stalled in the lower crust. There they assimilated some wall rock and some of this material was trapped in early crystallizing olivine. Then, higher in the crust, limestone assimilation buffered the Sr isotopic compositions.

Reply: As was shown above, the geochemistry of host rocks, constituent minerals and melt inclusions do not support the suggested preferred interpretation by Reviewer #3. It is impossible to accept that the earliest liquidus olivines would have trapped melt that was contaminated by any crustal source without leaving other noticeable traces in the geochemistry, mineralogy or textures of the rocks. Furthermore, it does not make sense to propose that a melt inclusion that was trapped within the lower crust can be buffered by limestone assimilation at higher levels in the crust: A melt inclusions that is trapped in primary olivine is inheritably shielded from processes that affects the composition of the magma during ascent to the surface.

The relations between isotope composition and trace elements require the involvement of an old component that had lost U to effectively freeze in the unradiogenic Pb composition and suggest mixing with small degree melts from this component to be trapped in early crystallising olivine, below the crust. Note that there is no evidence for the presence of old LCC basement within the crust beneath the Roman magmatic province (lines 227-229). Altogether, the geochemical relations and textures suggest recycling of LCC material (in addition to UCC), which is conceivable in the Italian post-collisional setting where multiple subduction systems and lithospheric delaminations have operated over tens of millions of years.

Whether the authors are willing to accept, or even consider, this interpretation is of course up to them, and to the editor.

We thank the reviewer for sharing his concerns. Although we do not accept the alternative interpretation, his comments prompted us to reexamine several of the points raised, which has led to improvements within the text as indicated.

References

- Armienti, P., Gasperini, D., 2007. Do We Really Need Mantle Components to Define Mantle Composition? *Journal of Petrology*, 48(4): 693-709.
- Avanzinelli, R., Lustrino, M., Mattei, M., Melluso, L., Conticelli, S., 2009. Potassic and ultrapotassic magmatism in the circum-Tyrrhenian region: Significance of carbonated pelitic vs. pelitic sediment recycling at destructive plate margins. *Lithos*, 113(1-2): 213-227.
- Boari, E., Tommasini, S., Laurenzi, M.A., Conticelli, S., 2009. Transition from Ultrapotassic Kamafugitic to Sub-alkaline Magmas: Sr, Nd, and Pb Isotope, Trace Element and ^{40}Ar - ^{39}Ar Age Data from the Middle Latin Valley Volcanic Field, Roman Magmatic Province, Central Italy. *Journal of Petrology*, 50(7): 1327-1357.
- Chapman, H.J., Moorbath, S., 1977. Lead isotope measurements from the oldest recognised Lewisian gneisses of north-west Scotland. *Nature*, 268(5615): 41-42.
- Conticelli, S., 1998. The effect of crustal contamination on ultrapotassic magmas with lamproitic affinity: mineralogical, geochemical and isotope data from the Torre Alfina lavas and xenoliths, Central Italy. *Chemical Geology*, 149(1): 51-81.
- Conticelli, S., Avanzinelli, R., Ammannati, E., Casalini, M., 2015. The role of carbon from recycled sediments in the origin of ultrapotassic igneous rocks in the Central Mediterranean. *Lithos*, 232: 174-196.
- Conticelli, S. et al., 2009a. Trace elements and Sr–Nd–Pb isotopes of K-rich, shoshonitic, and calc-alkaline magmatism of the Western Mediterranean Region: Genesis of ultrapotassic to calc-alkaline magmatic associations in a post-collisional geodynamic setting. *Lithos*, 107(1–2): 68-92.
- Conticelli, S. et al., 2009b. Shoshonite and sub-alkaline magmas from an ultrapotassic volcano: Sr–Nd–Pb isotope data on the Roccamonfina volcanic rocks, Roman Magmatic Province, Southern Italy. *Contributions to Mineralogy and Petrology*, 157(1): 41-63.
- Dickin, A.P., 1981. Isotope Geochemistry of Tertiary Igneous Rocks from the Isle of Skye, N.W. Scotland. *Journal of Petrology*, 22(2): 155-189.
- Downes, H., Dupuy, C., Leyrelop, A.F., 1990. Crustal evolution of the Hercynian belt of Western Europe: Evidence from lower-crustal granulitic xenoliths (French Massif Central). *Chemical Geology*, 83(3): 209-231.

- Downes, H., Kempton, P.D., Briot, D., Harmon, R.S., Leyreloup, A.F., 1991. Pb and O isotope systematics in granulite facies xenoliths, French Massif Central: implications for crustal processes. *Earth and Planetary Science Letters*, 102(3): 342-357.
- Frijia, G., Parente, M., Di Lucia, M., Mutti, M., 2015. Carbon and strontium isotope stratigraphy of the Upper Cretaceous (Cenomanian-Campanian) shallow-water carbonates of southern Italy: Chronostratigraphic calibration of larger foraminifera biostratigraphy. *Cretaceous Research*, 53: 110-139.
- Gaeta, M., Di Rocco, T., Freda, C., 2009. Carbonate Assimilation in Open Magmatic Systems: the Role of Melt-bearing Skarns and Cumulate-forming Processes. *Journal of Petrology*, 50(2): 361-385.
- Gaeta, M. et al., 2006. Time-dependent geochemistry of clinopyroxene from the Alban Hills (Central Italy): Clues to the source and evolution of ultrapotassic magmas. *Lithos*, 86(3): 330-346.
- Hawkesworth, C., Vollmer, R., 1979. CRUSTAL CONTAMINATION VERSUS ENRICHED MANTLE - ND-143-ND-144 AND SR-87-SR-86 EVIDENCE FROM THE ITALIAN VOLCANICS. *Contributions to Mineralogy and Petrology*, 69(2): 151-165.
- Iacono-Marziano, G., Scaillet, B., Gaillard, F., Pichavant, M., Chiodini, G., 2009. Role of non-mantle CO₂ in the dynamics of volcano degassing: The Mount Vesuvius example. *Geology*, 37(4): 319-322.
- Koornneef, J.M. et al., 2012. Melting of a Two-component Source beneath Iceland. *Journal of Petrology*, 53(1): 127-157.
- Langmuir, C.H., Vocke Jr, R.D., Hanson, G.N., Hart, S.R., 1978. A general mixing equation with applications to Icelandic basalts. *Earth and Planetary Science Letters*, 37(3): 380-392.
- Lustrino, M., 2005. How the delamination and detachment of lower crust can influence basaltic magmatism. *Earth-Science Reviews*, 72(1-2): 21-38.
- Lustrino, M. et al., 2016. Ca-rich carbonates associated with ultrabasic-ultramafic melts: Carbonatite or limestone xenoliths? A case study from the late Miocene Morron de Villamayor volcano (Calatrava Volcanic Field, central Spain). *Geochimica et Cosmochimica Acta*, 185: 477-497.
- Melluso, L., D'Antonio, M., Conticelli, S., Mirco, N.P., Saccani, E., 2003. Petrology and mineralogy of wollastonite- and melilite-bearing paralavas from the Central Apennines, Italy. *American Mineralogist*, 88(8-9): 1287-1299.
- Peccerillo, A., 1999. Multiple mantle metasomatism in central-southern Italy: Geochemical effects, timing and geodynamic implications. *Geology*, 27(4): 315-318.
- Peccerillo, A., 2017. *Cenozoic Volcanism in the Tyrrhenian Sea Region*. Springer International Publishing, Cham, 399 pp.
- Peccerillo, A., Federico, M., Barbieri, M., Brillì, M., Wu, T.-W., 2010. Interaction between ultrapotassic magmas and carbonate rocks: Evidence from geochemical and isotopic (Sr, Nd, O) compositions of granular lithic clasts from the Alban Hills Volcano, Central Italy. *Geochimica Et Cosmochimica Acta*, 74(10): 2999-3022.
- Peccerillo, A., Lustrino, M., 2005. Compositional variations of Plio-Quaternary magmatism in the circum-Tyrrhenian area: Deep versus shallow mantle processes. *Geological Society of America Special Papers*, 388: p. 421-434.
- Rogers, N.W., Hawkesworth, C.J., Parker, R.J., Marsh, J.S., 1985. The geochemistry of potassic lavas from Vulcini, central Italy and implications for mantle enrichment processes beneath the Roman region. *Contributions to Mineralogy and Petrology*, 90(2): 244-257.
- Rudge, J.F., Maclennan, J., Stracke, A., 2013. The geochemical consequences of mixing melts from a heterogeneous mantle. *Geochimica et Cosmochimica Acta*, 114: 112-143.
- Saal, A.E., Hart, S.R., Shimizu, N., Hauri, E.H., Layne, G.D., 1998. Pb isotopic variability in melt inclusions from oceanic island basalts, Polynesia. *Science*, 282(5393): 1481-1484.
- Stracke, A., Bizimis, M., Salters, V.J.M., 2003. Recycling oceanic crust: Quantitative constraints. *Geochemistry Geophysics Geosystems*, 4: 8003, doi:10.1029/2001GC000223. .

REVIEWERS' COMMENTS:

Reviewer #3 (Remarks to the Author):

I'm afraid I still cannot give a very positive evaluation. The main problem is that the authors have not adequately explained the most remarkable features of their results; 1) the contrast between the wide range of Pb isotope compositions with the conspicuously uniform Sr and (to a lesser extent) Nd isotope compositions and 2) the very unusual trace element compositions of the component with non-radiogenic Pb. To explain these results they propose that two components existed in the mantle beneath the Italian volcanoes, one with relatively radiogenic (high) Pb isotope ratios that they identify as "sediment-metasomatised mantle", the other with low ratios, but with essentially the same Sr and Nd isotope ratios, which they identify as lower continental crust that has found its way into the mantle. The 2nd component also has extremely high contents of moderately to highly incompatible trace elements. My problem is that I think it most unlikely that two components with these rather special characteristics exist in the mantle in such a form that they alone can be tapped without extraction of material with other compositions. It's true that many authors propose that magmas of uniform compositions can be extracted from the lithospheric mantle but I have always found this idea highly implausible. Particularly in a subduction setting, the mantle is likely to be highly heterogeneous with components with compositions ranging from that of depleted mantle, variably altered oceanic crust, both oceanic and continental sediments and perhaps material from the lower continental crust. The extent of variation probably resembles that of the volcanic rocks shown in the authors' Fig 3b, where $^{87}/^{86}\text{Sr}$ ranges from .702 to .718. The contrast between this large range and that of the melt inclusions (excluding two outliers .7096 to .7099) is striking. As the authors say, the mantle source probably consists of a network of veins and zones with compositions that vary, not only in terms of trace elements but also isotopically. Any process that extracts magma from this heterogeneous mantle source is likely to tap many of these components. I just cannot imagine that from this mantle zoo, just two components, coincidentally with the same, very restricted Sr isotope composition, could be tapped. It seems most unlikely that just these two components were tapped, with no discernible contribution from the presumably mantle source of the volcanic where $^{87}/^{86}\text{Sr}$ varies from .703 to .716.

The second problem is the unusual trace element content of the non-radiogenic components. Concentrations measured in the melt inclusions range from 10x higher than in upper continental crust and 10-30x higher than lower continental crust. It is also notable that the melt inclusions do not show depletion of U and Th relative to other trace elements, as is normal in lower crust with non-radiogenic Pb. The fact that the incompatible trace elements are higher in the "lower crustal component" than in "sediment-metasomatised mantle" is also puzzling. The authors discuss the possibility of increase in trace element contents during partial melting of the source but this implies that the percentage melting was less than 10% if the "exotic component" alone was tapped. On line 202 they propose that "carbonate-rich material was added to the unradiogenic LCC component after the ancient orogenic event". Did this component also have the same Sr and Nd isotope composition as the "sediment-metasomatised mantle" and the "lower crustal component". I readily admit that I can think of no easy explanation of the authors' results. For me, if the non-radiogenic component is indeed lower crust (and this is not proven and is inconsistent with the trace elements) then the most logical place for this component to have been added is in the crust. The authors state that the characteristics of the melt inclusions and host olivines point to a source in the mantle, but the characteristics they cite are not inconsistent with crystallization and melt entrapment from little-evolved magmas in a crustal setting. Is a primary magma with >12% MgO that initially crystallized Fo₉₂ inconceivable? To explain the high Sr content of the non-radiogenic component is tough but I still would look to assimilation of limestone or maybe evaporite, rather than stuff the source out of sight and mind somewhere in the mantle.

My recommendation therefore is that a paper which presents the melt inclusion data and highlights the weird characteristics of the two end-members could be published. There should of course be discussion of what these end-members could be, but the "lower-crust-in-the-mantle" interpretation should be discussed as one hypothesis to be evaluated together with sediment assimilation models. Given these worries, I think that the long section discussing how lower crust could have

got into the mantle is irrelevant.

Our replies to the remaining concerns of Reviewer #3:

I'm afraid I still cannot give a very positive evaluation. The main problem is that that authors have not adequately explained the most remarkable features of their results; 1) the contrast between the wide range of Pb isotope compositions with the conspicuously uniform Sr and (to a lesser extent) Nd isotope compositions and 2) the very unusual trace element compositions of the component with non-radiogenic Pb.

Reply: We have in our previous round of revisions explained both these points. We have presented a mixing model to explain the relations in isotope space and show that there are ancient LCC rocks in Europe (and Asia) that have adequate isotope compositions. We also explained that the trace element enrichment coupled to the isotope variability likely reflects mixing of small degree partial melts from the apparent two components. See lines 166-208

To explain these results they propose that two components existed in the mantle beneath the Italian volcanoes, one with relatively radiogenic (high) Pb isotope ratios that they identify as "sediment-metasomatised mantle", the other with low ratios, but with essentially the same Sr and Nd isotope ratios, which they identify as lower continental crust that has found its way into the mantle. The 2nd component also has extremely high contents of moderately to highly incompatible trace elements. My problem is that I think it most unlikely that two components with these rather special characteristics exist in the mantle in such a form that they alone can be tapped without extraction of material with other compositions. It's true that many authors propose that magmas of uniform compositions can be exacted from the lithospheric mantle but I have always found this idea highly implausible. Particularly in a subduction setting, the mantle is likely to be highly heterogeneous with components with compositions ranging from that of depleted mantle, variably altered oceanic crust, both oceanic and continental sediments and perhaps material from the lower continental crust. The extent of variation probably resembles that of the volcanic rocks shown in the authors' Fig 3b, where $^{87}/^{86}\text{Sr}$ ranges from .702 to .718. The contrast between this large range and that of the melt inclusions (excluding two outliers .7096 to .7099) is striking. As the authors say, the mantle source probably consists of a network of veins and zones with compositions that vary, not only in terms of trace elements but also isotopically. Any process that extracts magma from this heterogeneous mantle source is likely to tap many of these components. I just cannot imagine that from this mantle

zoo, just two components, coincidentally with the same, very restricted Sr isotope composition, could be tapped. It seems most unlikely that just these two components were tapped, with no discernible contribution from the presumably mantle source of the volcanic where $^{87}/^{86}\text{Sr}$ varies from .703 to .716.

Reply: We appreciate that the reviewer acknowledges that the mantle below Italy is expected to be highly heterogeneous “with components with compositions ranging from that of depleted mantle, variably altered oceanic crust, both oceanic and continental sediments and perhaps material from the lower continental crust”. However, we do not understand why the reviewer argues that our population of melt inclusions trapped in high Fo olivine extracted from 2 individual lava flows from 2 adjacent localities (Roccamonfina and Ernici) would be expected to record the full range of heterogeneity of the whole Italian arc. Note that these inclusions represent melts extracted from a certain depth and region beneath a relatively small area in Roman Magmatic Province. We do not expect the trapped melts to be representative of all melts generated along the entire subduction zone.

The second problem is the unusual trace element content of the non-radiogenic components. Concentrations measured in the melt inclusions range from 10x higher than in upper continental crust and 10-30x higher than lower continental crust. It is also notable that the melt inclusions do not show depletion of U and Th relative to other trace elements, as is normal in lower crust with non-radiogenic Pb. The fact that the incompatible trace elements are higher in the “lower crustal component” than in “sediment-metasomatised mantle” is also puzzling. The authors discuss the possibility of increase in trace element contents during partial melting of the source but this implies that the percentage melting was less than 10% if the “exotic component” alone was tapped. On line 202 they propose that “carbonate-rich material was added to the unradiogenic LCC component after the ancient orogenic event”. Did this component also have the same Sr and Nd isotope composition as the “sediment-metasomatised mantle” and the “lower crustal component”.

Reply: We indeed suggest that the trace element enriched nature of the MIs can result from mixing with small degree partial melts of the exotic source. Note that based on the combined major, trace element and isotope data the exotic source appears to be Ca-rich and to have a Sr (and Nd) isotope composition similar to the mantle modified by sediment addition. We do not distinguish between the compositions of the LCC and the potentially added Ca-rich material; the observed compositional variability suggests that it behaves as a single exotic component.

I readily admit that I can think of no easy explanation of the authors' results. For me, if the non-radiogenic component is indeed lower crust (and this is not proven and is inconsistent with the trace elements) then the most logical place for this component to have been added is in the crust. The author state that the characteristics of the melt inclusions and host olivines point to a source in the mantle, but the characteristic they cite are not inconsistent with crystallization and melt entrapment from little-evolved magmas in a crustal setting. Is a primary magma with >12% MgO that initially crystallized Fo92 inconceivable? To explain the high Sr content of the non-radiogenic component is tough but I still would look to assimilation of limestone or maybe evaporite, rather than stuff the source out of sight and mind somewhere in the mantle.

Reply: We refer to our extensive replies in the previous round of review. Assimilation within the crust can be excluded for our MI population in primitive (see Supplementary Note 1 p1-4). We are somewhat surprised that above the reviewer states that he expects the mantle to be highly

heterogeneous as a result of subduction recycling, but does not except that LCC material could reside in the mantle.

My recommendation therefore is that a paper which presents the melt inclusion data and highlights the weird characteristics of the two end-members could be published. There should of course be discussion of what these end-members could be, but the “lower-crust-in-the-mantle” interpretation should be discussed as one hypothesis to be evaluated together with sediment assimilation models. Given these worries, I think that the long section discussing how lower crust could have got into the mantle is irrelevant.

Reply: We thank the reviewer again for his suggestion to publish the data. As we do not agree with reviewer #3’s suggestion that crustal assimilation is relevant to the data-set, we choose to focus our discussion on the preferred model suggesting the role of a lower crust component in the mantle. Discussion of the potential influences of crustal assimilation is concentrated in the Supplementary Information. We remain of the opinion that a discussion of how a LLC component ended up in the mantle is of interest to a broad community of geochemists and geodynamicists.